# Multi-omic association study identifies DNA methylation-mediated genotype and smoking exposure effects on lung function in children living in urban settings

Matthew Dapas[1]*, Emma E. Thompson[1], William Wentworth-Sheilds[1], Selene Clay[1], Cynthia M. Visness[2], Agustin Calatroni[2], Joanne E. Sordillo[3], Diane R. Gold[4,5], Robert A. Wood[6], Melanie Makhija[7], Gurjit K. Khurana Hershey[8,9], Michael G. Sherenian[8,9], Rebecca S. Gruchalla[10], Michelle A. Gill[11], Andrew H. Liu[12], Haejin Kim[13], Meyer Kattan[14], Leonard B. Bacharier[15], Deepa Rastogi[16], Matthew C. Altman[17], William W. Busse[18], Patrice M. Becker[19], Dan Nicolae[20], George T. O'Connor[21], James E. Gern[18], Daniel J. Jackson[18], Carole Ober[1]*

**1** Department of Human Genetics, University of Chicago, Chicago Illinois, United States of America, **2** Rho Inc., Durham, North Carolina, United States of America, **3** Department of Population Medicine, Harvard Medical School, Boston, Massachusetts, United States of America, **4** Department of Environmental Health, Harvard T.H. Chan School of Public Health, Boston, Massachusetts, United States of America, **5** Channing Division of Network Medicine, Brigham and Women's Hospital, Harvard Medical School, Boston, Massachusetts, United States of America, **6** Department of Pediatrics, Johns Hopkins University Medical Center, Baltimore, Maryland, United States of America, **7** Division of Allergy and Immunology, Ann & Robert H. Lurie Children's Hospital, Chicago, Illinois, United States of America, **8** Department of Pediatrics, University of Cincinnati College of Medicine, Cincinnati, Ohio, United States of America, **9** Division of Asthma Research, Cincinnati Children's Hospital Medical Center, Cincinnati, Ohio, United States of America, **10** Department of Internal Medicine, University of Texas Southwestern Medical Center, Dallas, Texas, United States of America, **11** Department of Pediatrics, Washington University School of Medicine, St. Louis, Missouri, United States of America, **12** Department of Allergy and Immunology, Children's Hospital Colorado, University of Colorado School of Medicine, Aurora, Colorado, United States of America, **13** Department of Medicine, Henry Ford Health System, Detroit, Michigan, United States of America, **14** Columbia University College of Physicians and Surgeons, New York, New York, United States of America, **15** Monroe Carell Jr. Children's Hospital at Vanderbilt University Medical Center, Nashville, Tennessee, United States of America, **16** Children's National Health System, Washington, District of Columbia, United States of America, **17** Department of Allergy and Infectious Diseases, University of Washington, Seattle, Washington, United States of America, **18** Department of Pediatrics and Medicine, University of Wisconsin School of Medicine and Public Health, Madison, Wisconsin, United States of America, **19** National Institute of Allergy and Infectious Diseases, Bethesda, Maryland, United States of America, **20** Department of Statistics, University of Chicago, Chicago, Illinois, United States of America, **21** Pulmonary Center, Boston University School of Medicine, Boston, Massachusetts, United States of America

* mdapas@uchicago.edu (DM); c-ober@genetics.uchicago.edu (OC)

## Abstract

Impaired lung function in early life is associated with the subsequent development of chronic respiratory disease. Most genetic associations with lung function have been identified in adults of European descent and therefore may not represent those most relevant to pediatric populations and populations of different ancestries. In this study, we performed genome-wide association analyses of lung function in a multiethnic cohort of children (n = 1,035) living in low-income urban neighborhoods. We identified one novel locus at the *TDRD9* gene in chromosome 14q32.33 associated with percent predicted forced expiratory volume in one

**Data Availability Statement:** Phenotype, genotype, GWAS summary statistics, and whole-genome sequencing files are available in dbGaP

under accession phs002921.v1.p1. The URECA gene expression data are available on the Gene Expression Omnibus (GEO) under accession numbers GSE145505 (NECs) and GSE96783 (PBMCs). The URECA DNA methylation data are available in GEO under the reference series GSE217337. The pcHI-C data are available in GEO under the reference series GSE152550.

**Funding:** This work was supported by NIH grants U19 AI62310, HHSN272200900052C, HHSN272201000052I, UM1 AI114271, UG3 OD023282, and UM1 AI160040. Site data collection was supported by the following NIH grants: RR00052, UL1 TR001079 (Baltimore); M01 RR00533, UL1 RR025771, 1 UL1 TR001430 (Boston); UL1 TR000150 (Chicago); UL1 TR000451, UL1 TR001105 (Dallas); Ul1 RR025780 (Denver); UL1 TR000040, M01 RR00071, UL1 RR024156 (New York); UL1 TR000075 (D.C.); UL1 TR000077 (Cincinnati). M. D. was supported by TL1 TR002388 and T32 HL007605.

**Competing interests:** All authors, with the exception of M. Altman and P. Becker, report grants from NIH/NIAID during the conduct of study. M. Dapas, C. Visness, A. Calatroni and P. Becker have nothing to disclose outside the submitted work. C. Ober reports personal fees from American Association of Asthma, Allergy and Immunology, outside the submitted work. M. Altman reports personal fees from Sanofi-Regeneron outside the submitted work. W. Busse reports personal fees from Boston Scientific, Novartis, Glaxo SmithKline, Genentech, Sanofi/Genzyme, AstraZeneca, Teva, Regeneron and Elsevier outside the submitted work. M. Gill reports an honorarium for and support for travel to the 2017 AAAAI meeting during the conduct of study and monetary compensation from the American Academy of Pediatrics for her work teaching the biannual Pediatrics board review course, PREP The Course. K. Hershey reports grants from Adare, during the conduct of the study. D. Jackson reports personal fees from Novartis, Boehringer Ingelheim, Pfizer, Regeneron, AstraZeneca, Sanofi and Vifor Pharma, grants and personal fees from GlaxoSmithKline and grants from NIH/NHLBI, outside the submitted work. M. Kattan reports personal fees from Regeneron, outside the submitted work. R. Gruchalla reports government employment from Center for Biologics Evaluation and Research as well as personal fees from Consulting Massachusetts Medical Society, outside the submitted work. A. Liu reports personal fees from Phadia ThermoFisher as consulting honoraria, grants and non-financial support from

second ($FEV_1$) (p = $2.4 \times 10^{-9}$; $\beta_z$ = -0.31, 95% CI = -0.41- -0.21). Mendelian randomization and mediation analyses revealed that this genetic effect on $FEV_1$ was partially mediated by DNA methylation levels at this locus in airway epithelial cells, which were also associated with environmental tobacco smoke exposure (p = 0.015). Promoter-enhancer interactions in airway epithelial cells revealed chromatin interaction loops between $FEV_1$-associated variants in *TDRD9* and the promoter region of the *PPP1R13B* gene, a stimulator of p53-mediated apoptosis. Expression of *PPP1R13B* in airway epithelial cells was significantly associated the $FEV_1$ risk alleles (p = $1.3 \times 10^{-5}$; $\beta$ = 0.12, 95% CI = 0.06–0.17). These combined results highlight a potential novel mechanism for reduced lung function in urban youth resulting from both genetics and smoking exposure.

## Author summary

Lung function is determined by both genetic and environmental factors. Impairment of lung function can result from harmful environmental exposures in early life, which disproportionally affect children living in low-income, urban communities. However, most genetic association studies of lung function have been performed in adults and without regard for socioeconomic status. Therefore, genetic risk factors discovered to date may not reflect those most relevant to high-risk populations. In this study, we sought to identify genetic variants correlated with lung function in a multiethnic cohort of children living in low-income, urban neighborhoods and analyze how tobacco smoke exposure may influence any genetic effects. We discovered a common genetic variant associated with lower lung function in this population, and we found that the association was mediated by nearby epigenetic changes in DNA methylation, which were in turn correlated with smoking exposure. We then identified a nearby gene, *PPP1R13B*, which is known to aid in the deactivation of damaged cells, whose expression in airway cells aligned with these genetic and epigenetic effects. This study reveals a potential mechanism through which genetic risk and environmental exposures can affect airway development, perhaps leading to interventions that can help reduce the burden of asthma in socioeconomically disadvantaged children.

## Introduction

Reduced lung function is a hallmark of asthma and chronic obstructive pulmonary disease (COPD). Lung function measures, such as forced expiratory volume in one second ($FEV_1$) and forced vital capacity (FVC), are strong predictors of future all-cause mortality [1–6]. Airway obstruction often begins in early life [7–10], with lower lung function in infancy being a risk factor for the development of asthma in childhood [11] and COPD in late adulthood [12].

Genetic factors contribute to differences in lung function among individuals, with heritability estimates ranging from 0.50 for $FEV_1$ to 0.66 for $FEV_1$/FVC ratio [13]. The many genome-wide association studies (GWAS) of lung function measures [14–25] have implicated pathways related to lung development [20,26–28], inflammation [26], and tissue repair [29], among others [29]. Lung function is also affected by environmental exposures, such as smoking [30–32] and air pollution [33], which can disrupt airway development in early life, increasing the risk of childhood asthma and perhaps other chronic obstructive diseases [8–12,34,35]. For

ResMed/Propeller Health, non-financial support
from Revenio, grants and personal fees from
Avillion and personal fees from Labcorp, outside
the submitted work. L. Bacharier reports personal
fees from GlaxoSmithKline, Genentech/Novartis,
DBV Technologies, Teva, Boehringer Ingelheim,
AstraZeneca, WebMD/Medscape, Sanofi/
Regeneron, Vectura and geCircassia and personal
fees and non-financial support from Merckoutside
the submitted work. J. Gern reports personal fees
from AstraZeneca and Gossamer Bio and personal
fees and stock options from Meissa Vaccines Inc,
outside the submitted work. In addition, Dr. Gern
has a patent Methods of Propagating Rhinovirus C
in Previously Unsusceptible Cell Lines issued, and
a patent Adapted Rhinovirus C issued. G. O'Connor
reports personal fees from AstraZeneca and grants
from Janssen Pharmaceuticals, outside the
submitted work. R. Wood reports grants from
DBV, Aimmune, Astellas, HAL-Allergy and
Regeneron and royalties from Up to Date, outsite
the submitted work. Dr. Becker's co-authorship of
this publication does not necessarily constitute
endorsement by the National Institute of Allergy
and Infectious Diseases, the National Institutes of
Health or any other agency of the United States
government.

example, exposure to second hand smoke in utero and through childhood is associated with increased risk of childhood asthma [36], lower lung function in adolescence [37], and larger declines in lung function later in life [38,39]. Such adverse exposures are known to alter the epigenetic landscape in exposed individuals [40,41], potentially mediating downstream biological effects [42–44] and modifying genetic associations with lung function [45,46].

Environmental risk factors disproportionally affect socioeconomically disadvantaged children, particularly those living in urban environments [47,48]. In fact, socioeconomic effects contribute to disparities in lung health [49], including the higher burden of chronic respiratory disease among Black and Hispanic children compared to non-Hispanic white children [49–52]. Most genetic association studies of lung function, however, have been limited to adults of European descent. Therefore, genetic risk factors discovered to date may not reflect those most relevant to high-risk populations, which can further exacerbate health disparities [53,54]. Identifying genetic variants and epigenetic variation associated with lung function in high-risk, multiethnic, pediatric populations may provide more direct insights into the early development of impaired lung function.

In this study, we analyzed measures of lung function from the Asthma Phenotypes in the Inner City (APIC) [55,56] and Urban Environment and Childhood Asthma (URECA) cohorts [57], which consist of children living in low-income neighborhoods in 10 U.S. cities. We performed whole-genome sequencing (WGS) on 1,035 participants from APIC and URECA (ages 5–17 years; 67% non-Hispanic Black, 25% Hispanic; 66% with doctor-diagnosed asthma) and performed a GWAS with $FEV_1$ and the $FEV_1/FVC$ ratio. We then performed expression quantitative trait locus (eQTL) and methylation quantitative trait locus (meQTL) mapping in airway epithelial cells and peripheral blood mononuclear cells (PBMCs) from a subset of the URECA children. We further tested for genotype and DNA methylation interactions with smoking exposure. We aimed to identify methylation-mediated genetic and smoking exposure associations with lung function, linking environmental effects, epigenetic modifications, and specific genetic risk alleles to reduced pulmonary health in urban youth.

## Results

### Genetic variants at the *TDRD9* locus are associated with lung function

We completed WGS and variant calling on 1,035 participants from the APIC and URECA studies (APIC = 508, URECA = 527; Table 1). The mean sequencing depth was 31.6x per sample (S1A Fig). On average, 95.3%, 90.3% and 62.6% of each genome was mapped with at least 10x, 20x and 30x sequencing read depth, respectively (S1B Fig). Approximately 3.8 million high-confidence autosomal variants were called per sample. Variant call concordance between replicate sample pairs (n = 3) was >99.9% for single nucleotide polymorphisms (SNPs) and was 98.9% for insertions and deletions (InDels; S1 Table).

The sequenced cohort included 696 (67%) participants who self-identified as non-Hispanic Black and 258 (25%) who self-identified as Hispanic (Table 1). Principal component and admixture analyses using genotypes were conducted to characterize the ancestry of the participants (Fig 1). This revealed that the genetic ancestry of our sample was 66% African, 26% European, 7% Native American, and 1% East Asian. The cohort was 54% male and included 681 (66%) children diagnosed with asthma (Table 1).

Using the WGS variant calls for 14.1 million variants with minor allele frequency (MAF) ≥0.01, we performed a GWAS of two lung function traits: $FEV_1$ (% predicted) and $FEV_1/FVC$ (Z-scores), measured between ages 5–17 (Table 1, S2 Fig), adjusting for age, sex, asthma diagnosis, the first 10 principal components (PCs) of ancestry, and sample relatedness using a

**Table 1. Demographic characteristics of sequenced APIC and URECA participants.**

| Characteristic | All | APIC | URECA |
|---|---|---|---|
| Number | 1035 | 508 | 527 |
| Age, years, mean (SD) | 10.3 (2.5) | 10.9 (3.1) | 9.6 (1.1) |
| Female sex | 477 (46%) | 216 (43%) | 261 (50%) |
| *Race/Ethnicity* | | | |
| Black (non-Hispanic) | 696 (67%) | 319 (63%) | 377 (72%) |
| White (non-Hispanic) | 14 (1%) | 7 (1%) | 7 (1%) |
| Hispanic | 258 (25%) | 153 (30%) | 105 (20%) |
| Other/mixed | 64 (6%) | 26 (5%) | 38 (7%) |
| Unknown | 3 (<1%) | 2 (<1%) | 1 (<1%) |
| *Site* | | | |
| Baltimore | 234 (23%) | 85 (17%) | 149 (28%) |
| Boston | 189 (18%) | 65 (13%) | 124 (23%) |
| Chicago | 62 (6%) | 62 (12%) | - |
| Cincinnati | 45 (4%) | 45 (9%) | - |
| Dallas | 38 (4%) | 38 (9%) | - |
| Denver | 59 (6%) | 59 (12%) | - |
| Detroit | 50 (5%) | 50 (10%) | - |
| New York | 164 (16%) | 64 (13%) | 100 (19%) |
| St. Louis | 155 (15%) | - | 155 (29%) |
| Washington, D.C. | 39 (4%) | 39 (8%) | - |
| Household income < $15k | 550 (54%) | 273 (54%) | 277 (54%) |
| Caretaker completed HS | 756 (73%) | 364 (72%) | 392 (74%) |
| Caretaker smokes* | 315 (33%) | 123 (27%) | 192 (39%) |
| Asthma | 681 (75%) | 508 (100%) | 173 (43%) |
| BMI, Z-score, mean (SD) | 0.9 (1.2) | 1.0 (1.2) | 0.8 (1.1) |
| $FEV_1$, % predicted, mean (SD) | 94.9 (16.3) | 91.9 (17.6) | 98.5 (14.5) |
| FEV/FVC, mean (SD) | 0.80 (0.09) | 0.78 (0.10) | 0.83 (0.07) |

Results are presented as counts and percentages or as means with standard deviations. Missing data were not included in percentage calculations. Ages for URECA correspond to the year the genome-wide association study lung function data were collected.

*Caretaker smoking status in URECA was collected at age 10. APIC: Asthma Phenotypes in the Inner City; BMI: body mass index; $FEV_1$: forced expiratory volume in one second; $FEV_1/FVC$: ratio of $FEV_1$ to forced vital capacity; HS: high school; URECA: Urban Environment and Childhood Asthma.

linear mixed model [58]. The $FEV_1$ GWAS included 896 participants from APIC (n = 504) and URECA (n = 392), and the $FEV_1$/FVC GWAS included 886 participants from APIC (n = 497) and URECA (n = 389). The genomic control factor, $\lambda_{GC}$, for both GWAS results was 1.02 (S3 Fig), indicating adequate control for population stratification. We identified one locus on chromosome 14q32.33 that was associated with $FEV_1$ at genome-wide significance ($p<2.5x10^{-8}$); no other variants were associated with $FEV_1$ and no variants were associated with $FEV_1$/FVC at genome-wide levels of significance (Fig 2). The $FEV_1$ locus on chromosome 14 consisted of a 200 kb region of associated variants in high linkage disequilibrium (LD) across the *TDRD9* (Tudor Domain Containing 9) gene (Fig 3, S2 Table). The minor allele at the lead SNP (rs10220464; MAF = 0.30) was significantly associated with lower $FEV_1$ ($p = 2.4x10^{-9}$; $\beta_z = -0.31$, 95% confidence interval (CI) = -0.41- -0.21) and nominally associated

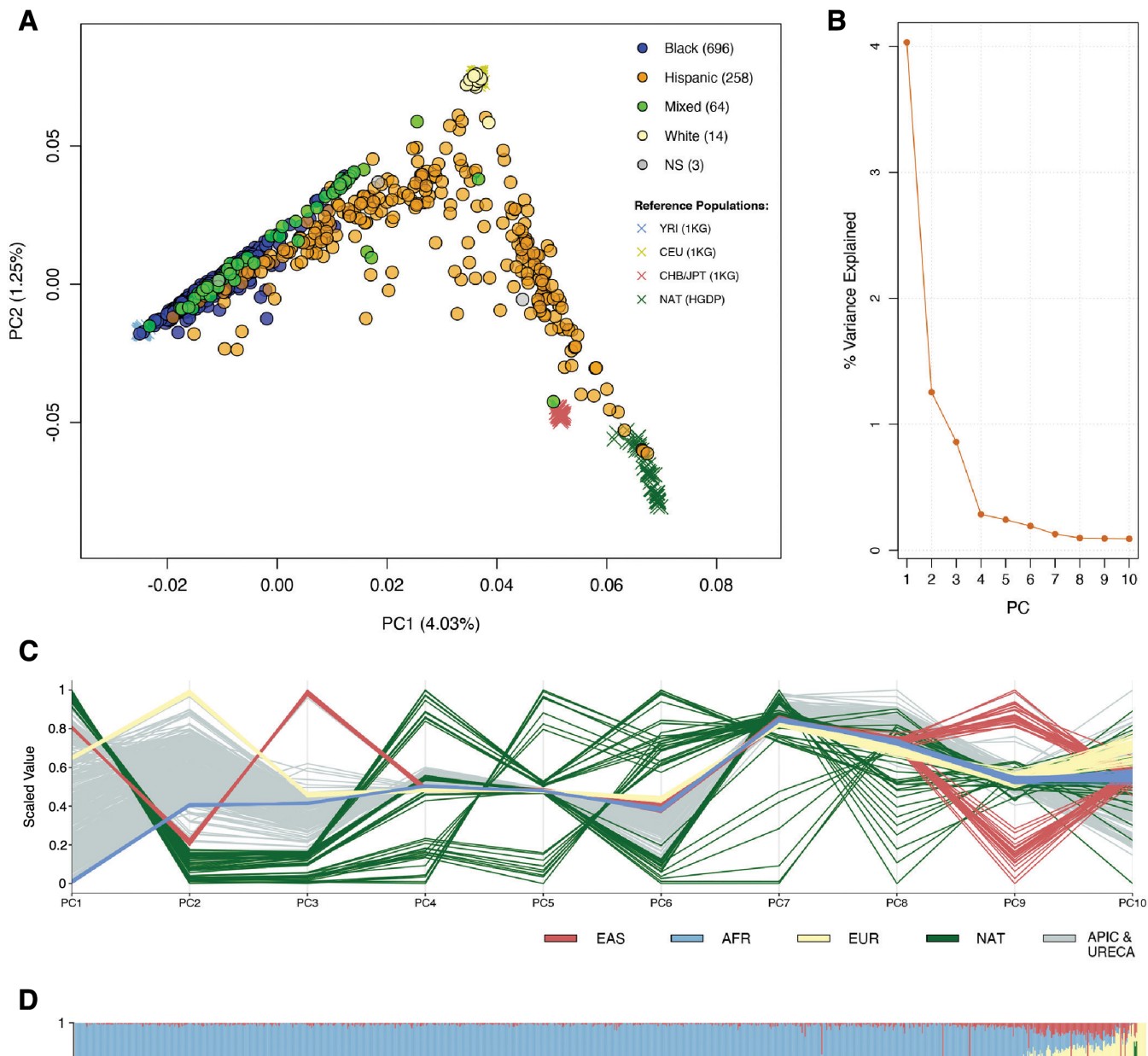

**Fig 1. Ancestry composition of sequenced APIC & URECA participants.** A) The top two principal components (PCs) of ancestry are plotted for sequenced APIC & URECA participants, colored by self-identified race/ethnicity, along with the four ancestry reference populations used for determining ancestry. NS = not specified. B) The proportion of genetic variance explained by each of the top 10 PCs. C) The relative values of the top 10 PCs are plotted for each sample, colored by reference population. D) The estimated proportion of admixture from each ancestral population is shown for each sequenced APIC & URECA participant. Each vertical line corresponds to one sample. 1KG, 1000 Genomes project; HGDP, Human Genome Diversity Project; YRI, Yoruba in Ibadan, Nigeria; CEU, Utah residents with Northern and Western European ancestry; CHB, Han Chinese in Beijing, China; JPT, Japanese in Tokyo, Japan; NAT, Native Americans from HGDP; EAS, East Asian ancestry; AFR, African ancestry; EUR, European ancestry.

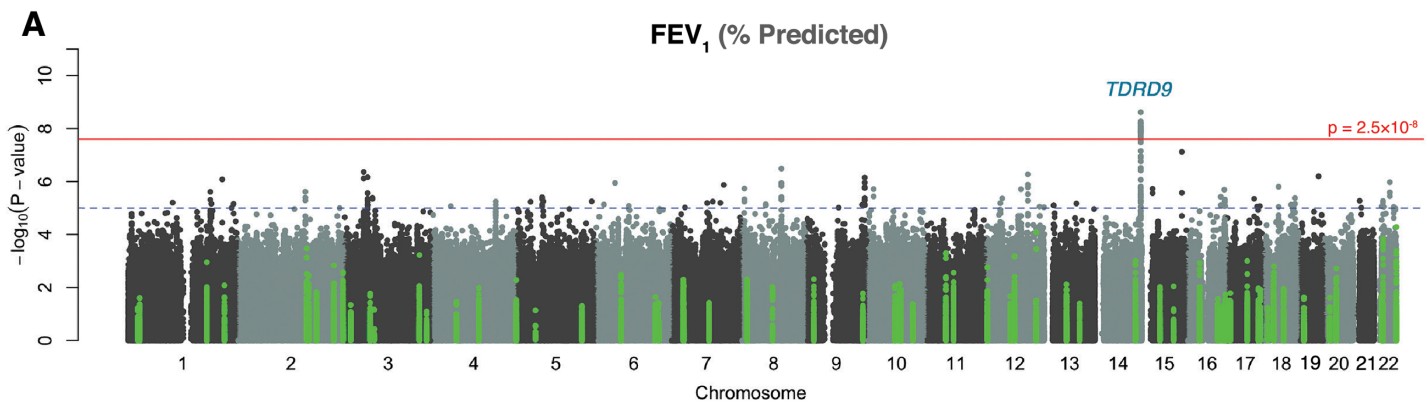

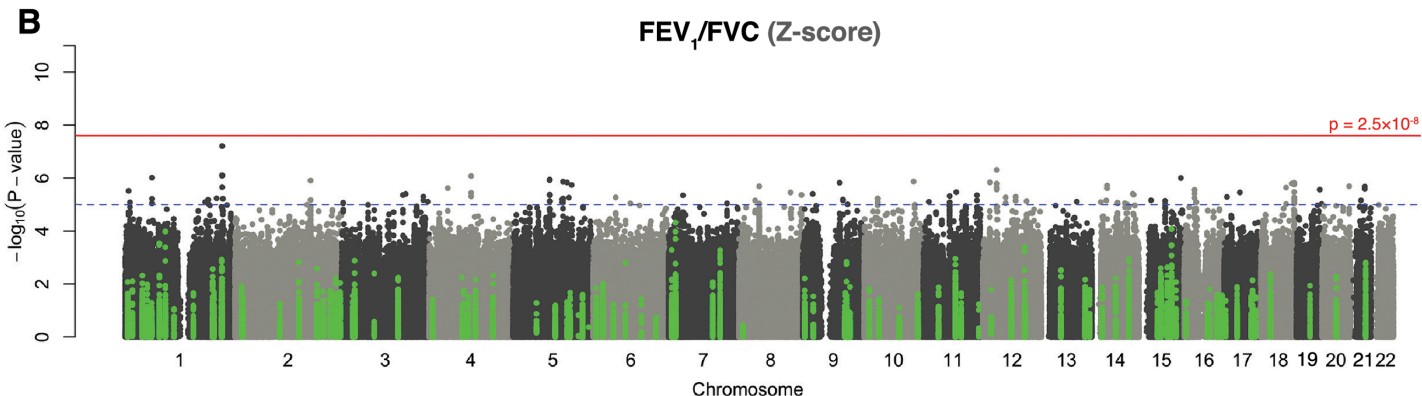

**Fig 2. Genome-wide association results.** GWAS Manhattan plots for **A)** FEV$_1$ and **B)** FEV$_1$/FVC ratio. The horizontal red line indicates genome-wide significance (p $\leq$ 2.5x10$^{-8}$). The dotted horizontal blue line indicates p = 1x10$^{-5}$. Variants colored in green are in previously identified GWAS loci [23]. FEV$_1$, forced expiratory volume in one second; FEV$_1$/FVC, ratio of FEV$_1$ to forced vital capacity.

with lower FEV$_1$/FVC (p = 1.1x10$^{-3}$; $\beta_z$ = -0.17, 95% CI = -0.28- -0.07). Fine-mapping analysis at this locus (chr14:103.7–104.3Mb) revealed one 95% credible set of effect variables consisting of 59 SNPs, with rs10220464 having the highest individual posterior inclusion probability among them (S4 Fig). We did not detect any significant differences in rs10220464 association effect size by ancestry or asthma status or study for FEV$_1$ (Fig 4). Furthermore, the *TDRD9* locus remained the only genome-wide significant association when the two GWAS were performed without adjustment for asthma status (S5 Fig). The overall effect size correlations between asthma-adjusted and unadjusted GWAS results were r = 0.981 for FEV$_1$ and r = 0.954 for FEV$_1$/FVC.

We examined association results for the previously identified FEV$_1$ and FEV$_1$/FVC loci reported in the meta-analysis of the UK Biobank and SpiroMeta Consortium by Shrine and colleagues (n = 400,102) [23], which included 70 loci for FEV$_1$ and 117 for FEV$_1$/FVC. Of these, 64 of the lead SNPs for FEV$_1$ and 112 for FEV$_1$/FVC were genotyped in the APIC and URECA sample. Only one SNP, for FEV$_1$, replicated with false discovery rate (FDR) q<0.05 (rs9610955; p = 1.0x10$^{-4}$; $\beta_z$ = -0.38, 95% CI = -0.58- -0.19; S6 and S7 Figs). Cumulatively, 56% (n = 36) and 54% (n = 60) of these SNPs demonstrated consistent directions of effect for FEV$_1$ and FEV$_1$/FVC, respectively, with effect size correlations of 0.29 (95% CI = 0.05–0.50; p = 0.020) for FEV$_1$ and 0.42 (95% CI = 0.25–0.56; p = 4.2x10$^{-6}$) for FEV$_1$/FVC.

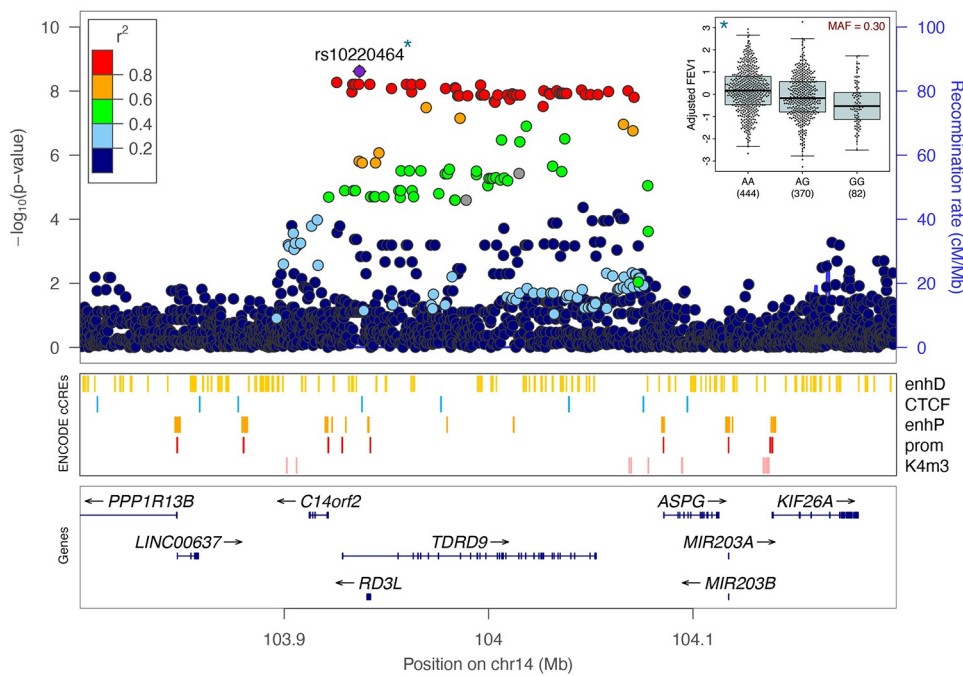

**Fig 3. FEV₁-associated variants on chromosome 14q32.33.** FEV$_1$ association results are shown at the *TDRD9* gene locus. Each variant is plotted according to its position and -log$_{10}$ p-value, colored by linkage disequilibrium to the lead variant, rs10220464, within the sample. Candidate cis-Regulatory Elements (cCREs) from ENCODE [59] are also shown for the region. The inset panel in the upper right shows the distribution of adjusted FEV$_1$ values by rs10220464 genotype. FEV$_1$, forced expiratory volume in one second; MAF, minor allele frequency; EnhD, distal enhancer-like signature; CTCF, CCCTC-binding factor sites; enhP, proximal enhancer-like signature; prom, promoter-like signature; K4m3, trimethylation of histone H3 at lysine 4.

## Lung function risk alleles are associated with DNA methylation at the *TDRD9* locus in airway epithelial cells

The majority of complex trait-associated variants exert effects by altering gene regulatory networks [60–62]. These changes are often marked by quantitative differences in DNA methylation levels [63–65]. We therefore investigated correlations between the FEV$_1$-associated allele at *TDRD9* and DNA methylation at the locus in upper airway (nasal) epithelial cells (NECs) from URECA children at age 11 (n = 286). We tested for associations between the FEV$_1$ genotype, as tagged by rs10220464, and DNA methylation levels at 796 CpG sites within 10 kb of any *TDRD9* locus variants associated with FEV$_1$ at p<1x10$^{-5}$ (n = 82 variants). The rs10220464 genotype was an meQTL for 5 CpG sites at an FDR <0.05 (S3 Table). DNA methylation levels at only one of these CpG sites, cg03306306 (p = 2.3 x10$^{-4}$; β = 0.07, 95% CI = 0.03–0.10; Fig 5A), was also significantly associated with FEV$_1$ at age 10 in URECA (p = 0.011; β = -11.48, 95% CI = -20.27- -2.69; Fig 5B). The rs10220464 genotype accounted for 4.7% of residual variation in cg03306306 methylation, and cg03306306 methylation explained 2.4% of residual variation in FEV$_1$.

We then analyzed cg03306306 methylation in PBMCs collected at age 7 (n = 169) [66] from URECA children to evaluate whether the genotype and lung function associations observed in NECs were shared with blood cells. In PBMCs, we observed no correlation between the rs10220464 risk allele and cg03306306 methylation (Fig 5C), nor was there an association between cg03306306 methylation and FEV$_1$ (Fig 5D). These results indicate that cg03306306 methylation dynamics in the airway epithelium are not present in peripheral blood cells.

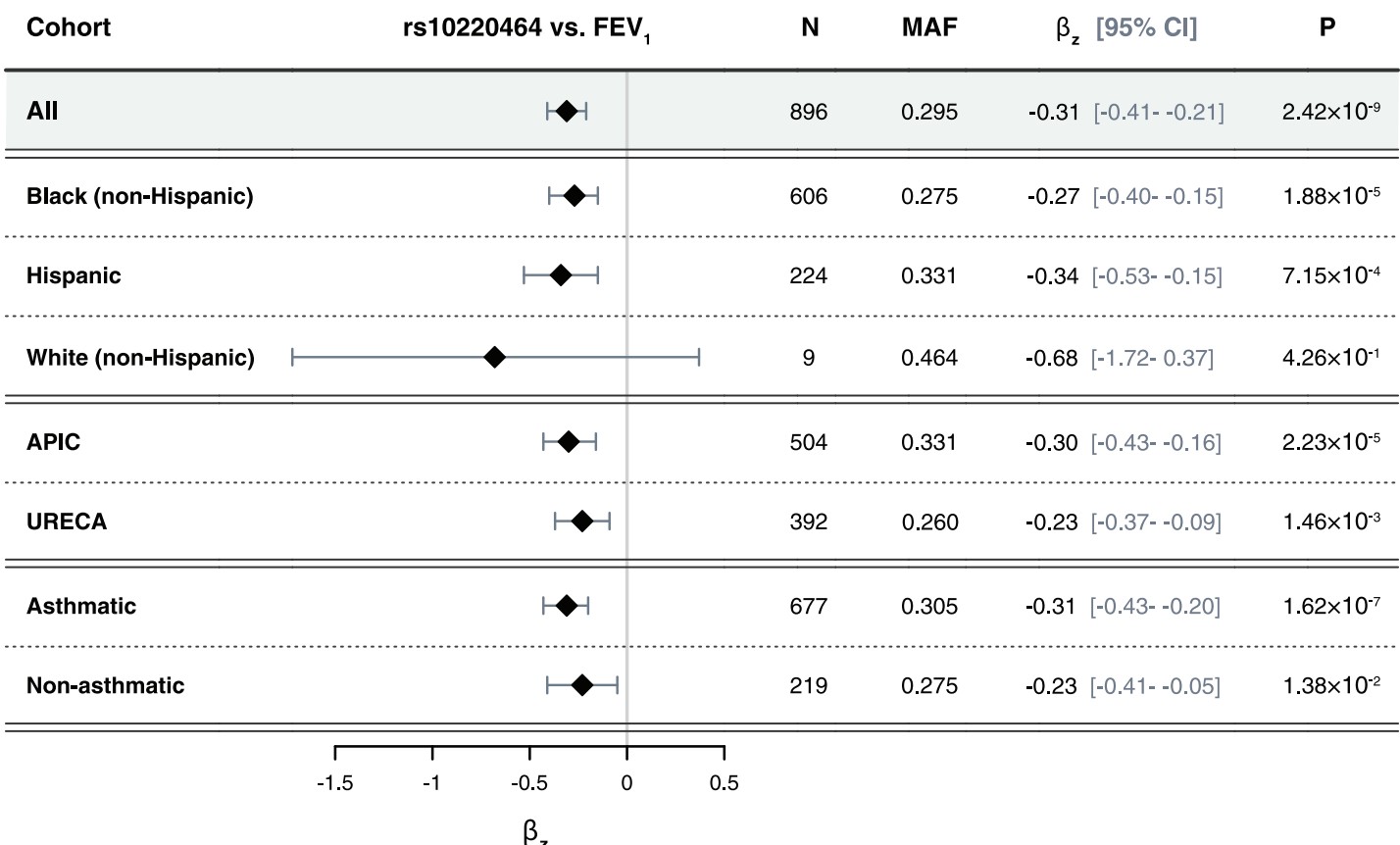

| Cohort | rs10220464 vs. FEV$_1$ | N | MAF | β$_z$ [95% CI] | P |
|---|---|---|---|---|---|
| **All** | | 896 | 0.295 | -0.31 [-0.41- -0.21] | $2.42×10^{-9}$ |
| **Black (non-Hispanic)** | | 606 | 0.275 | -0.27 [-0.40- -0.15] | $1.88×10^{-5}$ |
| **Hispanic** | | 224 | 0.331 | -0.34 [-0.53- -0.15] | $7.15×10^{-4}$ |
| **White (non-Hispanic)** | | 9 | 0.464 | -0.68 [-1.72- 0.37] | $4.26×10^{-1}$ |
| **APIC** | | 504 | 0.331 | -0.30 [-0.43- -0.16] | $2.23×10^{-5}$ |
| **URECA** | | 392 | 0.260 | -0.23 [-0.37- -0.09] | $1.46×10^{-3}$ |
| **Asthmatic** | | 677 | 0.305 | -0.31 [-0.43- -0.20] | $1.62×10^{-7}$ |
| **Non-asthmatic** | | 219 | 0.275 | -0.23 [-0.41- -0.05] | $1.38×10^{-2}$ |

**Fig 4. Rs10220464 effect size heterogeneity.** A forest plot of the associations between rs10220464 and FEV$_1$ (% predicted) are shown for distinct sub-cohorts distinguished by self-identified race/ethnicity, study, and asthma status. β$_z$, the association effect size between the rs10220464 allele count and the adjusted and normalized FEV$_1$ (% predicted) values; FEV$_1$, forced expiratory volume in one second; N, total number of individuals included in the association test; MAF, minor allele frequency within the sub-cohort; P, the association p-value.

## Smoking exposure is associated with DNA methylation at the *TDRD9* locus

DNA methylation at the *TDRD9* locus had previously been associated with maternal smoking during pregnancy [67,68]. Therefore, we tested for associations between environmental tobacco smoke exposure (S8 Fig) and DNA methylation at this locus in the URECA children. Methylation at cg03306306 in NECs was significantly associated with nicotine metabolite (cotinine) levels in urine collected at ages 7–10 years (p = 0.015; β = 0.03, 95% CI = 0.01–0.05; Fig 6). Methylation at cg03306306 in PBMCs from age 7 was not associated with urine cotinine levels.

To determine if there was an interaction effect between genotype and smoking exposure on DNA methylation and/or lung function, we repeated the cotinine association tests in URECA with the addition of an interaction term to assess if the genotype effect differed between individuals with low and high exposures to smoking. There were no significant genotype-by-smoking exposure interaction effects on methylation levels in NECs or PBMCs in URECA, nor were there any significant methylation-by-smoking effects on FEV$_1$ (S9 Fig). There was modest evidence for a genotype-by-smoking exposure interaction effect on FEV$_1$ in the combined APIC and URECA sample, but this did not reach statistical significance (p = 0.06, S10 Fig). Considering the ages of the participants in APIC and URECA, most tobacco exposures were likely due to secondhand smoke.

## NECs

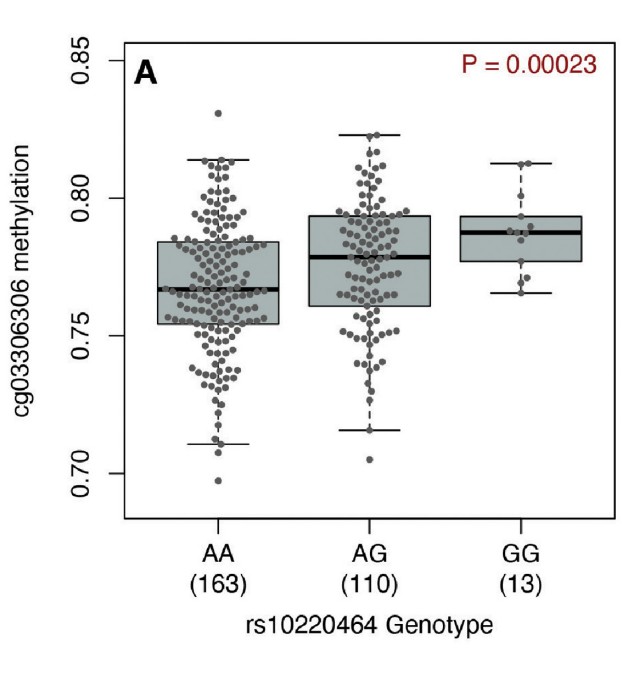

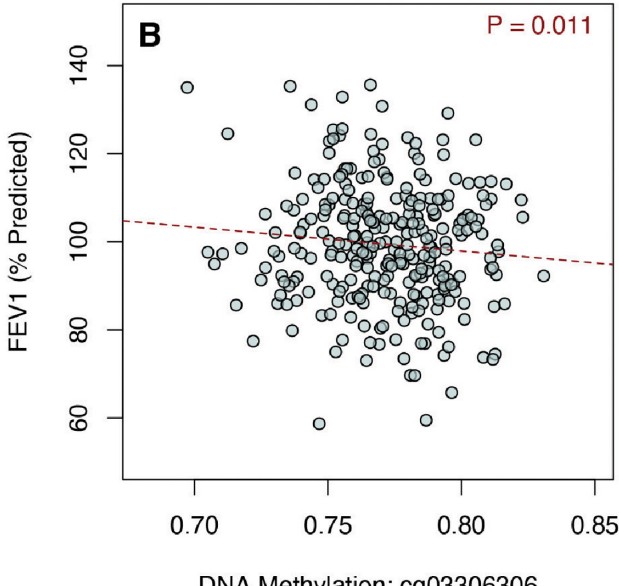

## PBMCs

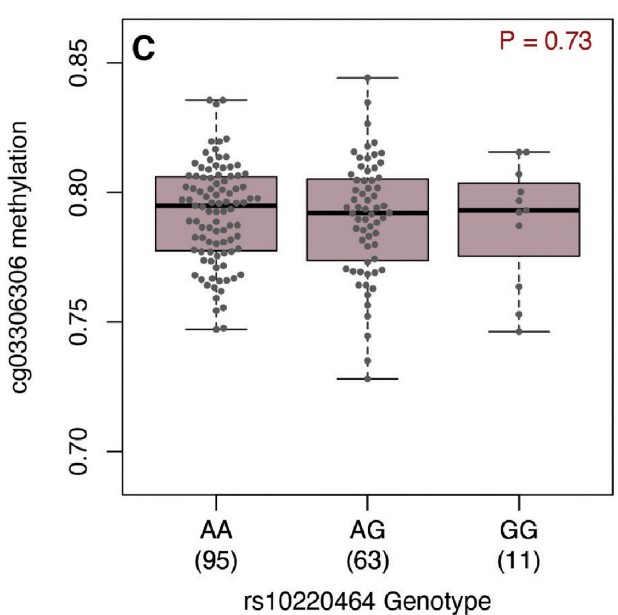

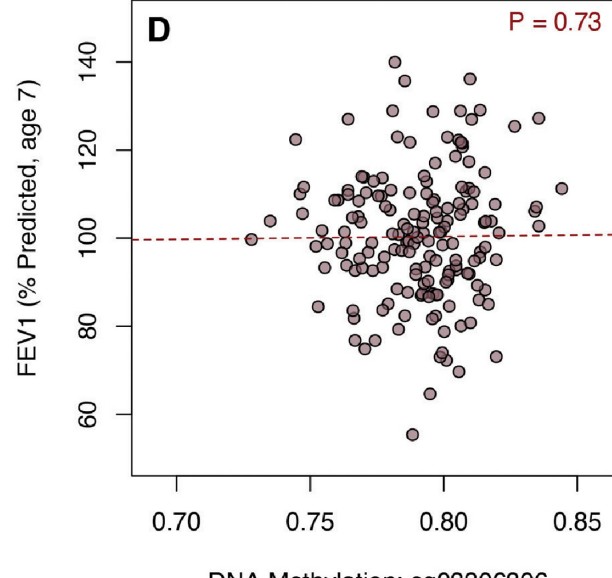

**Fig 5. Genotype and FEV₁ associations with DNA methylation.** DNA methylation levels at cg03306306 are shown by rs10220464 genotype and FEV$_1$ measures are plotted against cg03306306 methylation levels in NECs at age 11 (A, B), and PBMCs at age 7 (C, D) from URECA. FEV$_1$, forced expiratory volume in one second; NECs, nasal epithelial cells; PBMCs, peripheral blood mononuclear cells; URECA, Urban Environment and Childhood Asthma study.

### Genetic effects on lung function are mediated by DNA methylation

To determine if DNA methylation at the *TDRD9* locus had a causal effect on lung function, we performed a Mendelian randomization analysis using two-stage least squares (2SLS)

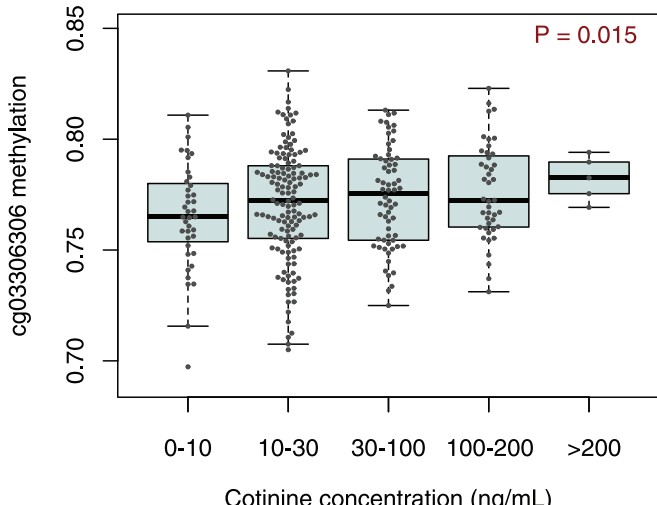

**Fig 6. DNA methylation association with smoking exposure.** DNA methylation at cg03306306 in nasal epithelial cells at age 11 are plotted against urine cotinine levels from URECA at ages 8–10 as measured using the NicAlert assay (n = 285). URECA, Urban Environment and Childhood Asthma study.

regression. In the first stage, cg03306306 methylation levels in NECs were regressed on an instrument composed of four meQTLs for cg03306306 (rs11160777, rs137961671, rs7143936, rs11160776; Materials and methods). In the second stage, $FEV_1$ was regressed on the predicted DNA methylation values generated from the first stage regression, thereby yielding a causal effect estimate of cg03306306 methylation on $FEV_1$. Urine cotinine levels were included as a covariate in both stages. The variance explained in the first stage regression was $r^2 = 0.11$. The causal effect of cg03306306 methylation on $FEV_1$ was statistically significant (p = 0.020). We also tested a single, unweighted allele score of the instrumental variables and observed a causal effect association of p = 0.045 (stage-one $r^2 = 0.10$). We further performed a bootstrapped mediation analysis to test whether the rs10220464 risk allele effect on $FEV_1$ was mediated by DNA methylation. The indirect effect of rs10220464 on $FEV_1$ via cg03306306 methylation was significant, both when including asthma as a covariate ($\beta_z$ = -0.04, 95% CI = -0.10- -0.003, percent mediated = 14.4%) and when asthma was not considered ($\beta_z$ = -0.04, 95% CI = -0.10- -0.002, percent mediated = 15.0%). These results indicate that the effect of the $FEV_1$-associated genotype at the *TDRD9* locus is partially mediated through its impact on nearby DNA methylation levels.

## Gene expression and promoter-enhancer interactions implicate *PPP1R13B*

Trait-associated variants and DNA methylation often affect the transcriptome by influencing the expression of one or more neighboring genes [69,70]. Identifying these correlations can help infer causal mechanisms [71]. Therefore, we next explored the relationship between the genotype for the lead $FEV_1$ variant rs10220464 and the expression of genes within 1 Mb in NECs and PBMCs from URECA children. Notably, the rs10220464 genotype was not associated with *TDRD9* expression levels in these cells (NECs: p = 0.60, β = 0.12; PBMCs: p = 0.91, β = 0.014). Of the 27 genes that were evaluated (S4 Table), rs10220464 was significantly associated with the expression of only one gene, *PPP1R13B* (Protein Phosphatase 1 Regulatory Subunit 13B; FDR q = $2.77.\times10^{-4}$; p = $1.3\times10^{-5}$; β = 0.12, 95% CI = 0.06–0.17; Fig 7A), in NECs. *PPP1R13B* expression levels were also the most strongly associated of the 27 genes with

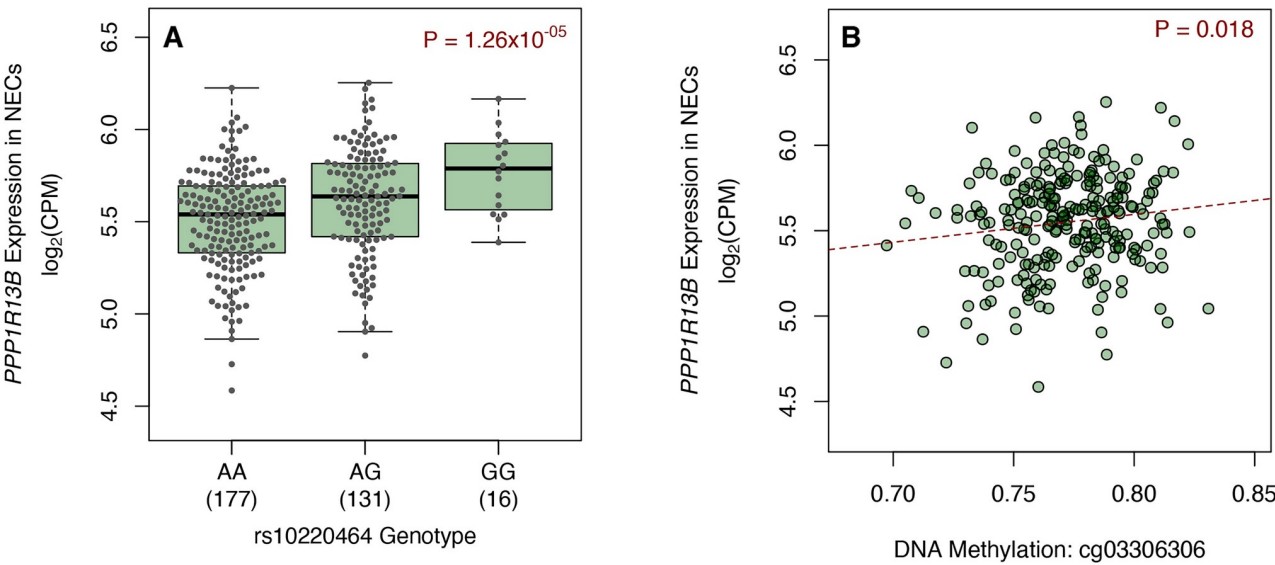

**Fig 7. *PPP1R13B* gene expression in NECs.** *PPP1R13B* gene expression in NECs at age 11 are plotted against A) rs10220464 genotype (n = 324) and B) DNA methylation at cg03306306 in NECs at age 11 (n = 254). NECs, nasal epithelial cells; CPM, counts per million.

methylation at cg03306306 in NECs (p = 0.018; β = 0.10, 95% CI = 0.02–0.18; Fig 7B). *PPP1R13B* expression in NECs, however, was not associated with $FEV_1$ or smoking exposure (S11 Fig).

The transcription start site of *PPP1R13B* resides 87 kb from rs10220464 and 152 kb from cg03306306, suggesting long-range interactions between the $FEV_1$-associated genotype and the promoter of *PPP1R13B*. To determine whether any of the $FEV_1$-associated GWAS variants at the *TDRD9* locus resided in regions that physically interacted with the promoters of *cis*-genes, we evaluated chromatin interactions in lower airway (bronchial) epithelial cells (BECs) [72], assessed by promoter-capture Hi-C. Forty-two of the GWAS variants resided in regions that interacted with the promoters of 9 different genes expressed in NECs (Fig 8; S5 Table). The gene most frequently mapped to these variants was *PPP1R13B*, with 15 variants located in 3 different interaction loops. Moreover, the strongest observed interaction was between a region containing 4 $FEV_1$-associated variants and the *PPP1R13B* promoter (CHiCAGO score = 9.38; S5 Table), suggesting that this region is an enhancer for *PPP1R13B* expression. This putative enhancer region is located just 2.21 kb from cg03306306.

## Summary of study associations

The associations between the *TDRD9* risk allele, cg03306306 DNA methylation in NECs, smoking exposure, *PPP1R13B* gene expression, and $FEV_1$ (% predicted) reported in this study are summarized in Fig 9.

## Discussion

Using whole-genome sequence variant calls in an asthma-enriched cohort of predominantly African-American children raised in urban environments, we identified a genotype at the *TDRD9* locus associated with lower $FEV_1$% predicted. This genotype effect was partially mediated by DNA methylation in airway epithelial cells, which were also correlated with smoking exposure. Data from RNA-sequencing and promoter-capture Hi-C in airway epithelial cells

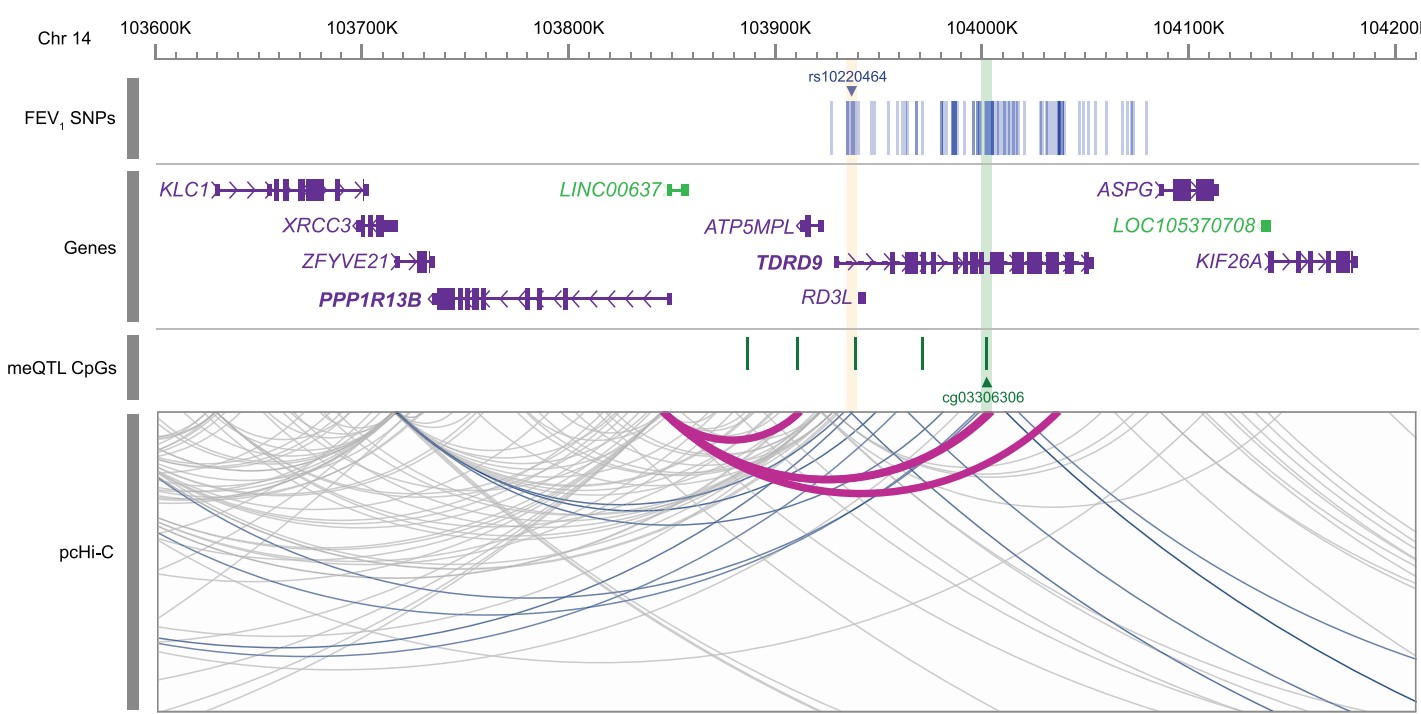

**Fig 8. Promoter-enhancer interactions at *TDRD9* locus in nasal epithelial cells.** Promoter-to-enhancer chromatin interactions captured by Hi-C in nasal epithelial cells from URECA at age 11 are displayed as grey arcs. SNPs associated with $FEV_1$ (p<1x10$^{-5}$) are marked by blue lines in the top row according to their genomic position on chromosome 14. The lead $FEV_1$ SNP, rs1022464, is highlighted in yellow. CpG sites associated with rs1022464 (FDR<0.05) are displayed as green markers below the genes, with cg03306306 highlighted in green. Chromatin Interactions containing SNPs associated with $FEV_1$ (p<1x10$^{-5}$) are highlighted in blue. Magenta arcs highlight interactions between the *PPP1R13B* promoter and regions containing $FEV_1$ SNPs and/or rs1022464-associated CpG sites. $FEV_1$, forced expiratory volume in one second; SNPs, single nucleotide polymorphisms; meQTL, methylation quantitative trait locus; pcHi-C, promoter capture Hi-C.

suggested that these $FEV_1$-associated genetic and epigenetic variations influence the expression of the *PPP1R13B* gene through long-range interactions.

The *PPP1R13B* gene encodes a protein that promotes apoptosis, a form of programmed cell death, via its interaction with the tumor suppressor p53 and is often referred to by its alias ASPP1 (apoptosis-stimulating protein of p53 1) [73]. In response to oncogenic stress, PPP1R13B translocates to the nucleus, where it enhances the transcriptional activity of p53 on specific target genes relevant to apoptosis [74,75]. Exposure to smoking and fine particulate matter induces epithelial apoptosis in the lung via p53 [76–78]. PPP1R13B may also promote apoptosis in a p53-independent manner by inhibiting autophagy in response to upregulation by EGR-1 (early growth response protein 1) [79]. EGR-1 mediates stress-induced proinflammatory responses in the airway epithelium and contributes to the pathogenesis of COPD [80–85]. Within the lung, *PPP1R13B* is indeed predominantly expressed in epithelial cells, particularly in alveolar type 2 cells, and less so in immune cells and fibroblasts [86,87]. However, Cheng and colleagues studied PPP1R13B function in lung fibroblasts and found that it was upregulated following $SiO_2$ exposure, where it promoted fibroblast proliferation and migration through endoplasmic reticulum stress and autophagy pathways [88]. Overall, these studies suggest that *PPP1R13B* plays a key role in maintaining tissue homeostasis by regulating apoptosis and autophagy in response to environmental stimuli [74,89,90]. The specific function(s) of this gene in the airway epithelium and its potential impact on the development of airway obstruction remain to be elucidated. *PPP1R13B* expression in airway epithelial cells at age 11 was not associated with lung function or urine cotinine levels in the URECA children, but the cofactors

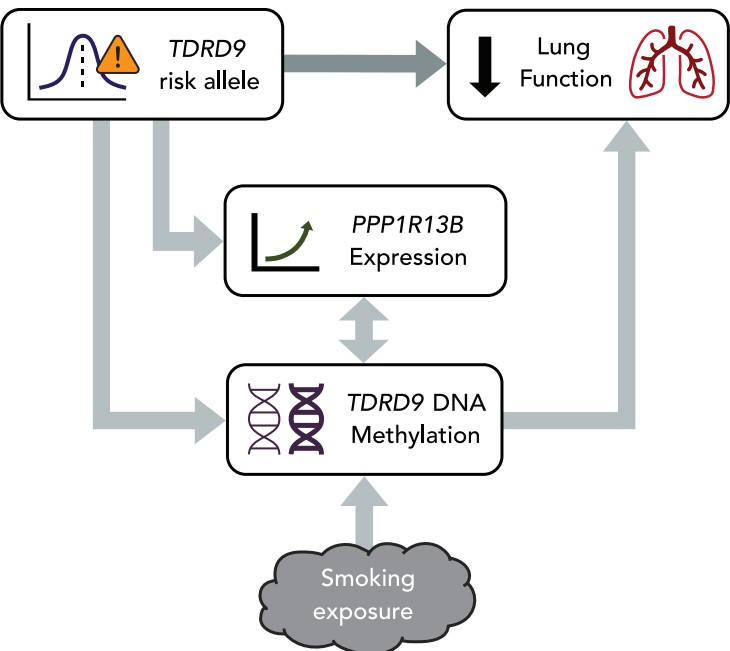

**Fig 9. Summary of study associations.** The *TDRD9* locus was significantly associated with FEV$_1$ (% predicted) in the APIC and URECA cohorts. This association was partially mediated by DNA methylation at the cg03306306 CpG site in *TDRD9* in NECs, which was also significantly associated with environmental tobacco smoke exposure. The TDRD9 risk allele and DNA methylation were both significantly associated with *PPP1R13B* gene expression, but *PPP1R13B* gene expression was not significantly correlated with FEV$_1$ itself. Unidirectional arrows represent inferred causality.

of this gene [79,91] have been found previously to be upregulated in smokers with COPD [81,92]. Given its association with lung function alleles in our study, its expression in the airway epithelium, and its purported functions in autophagy and apoptosis pathways, additional study of *PPP1R13B* in lung and airway development is warranted, particularly in the context of adverse environmental stimuli, many of which are enriched in low-income urban environments.

In NECs, *PPP1R13B* gene expression was significantly associated with DNA methylation levels at the cg03306306 CpG site in *TDRD9*. Methylation at the *TDRD9* locus was previously reported to correlate with specific environmental exposures [67,68,93] and with *TDRD9* expression in blood [67,94]. *TDRD9* is lowly expressed in the lung but is detected in alveolar macrophages and in monocytes [86,87]. Interestingly, the gene was among the most differentially expressed genes in alveolar macrophages in smokers relative to non-smokers [95], and its knockdown in TDRD9-expressing lung carcinomas resulted in increased apoptosis [96]. Its expression was not correlated with the rs10220464 genotype in URECA NECs or PBMCs, but rs10220464 is an eQTL for *TDRD9* expression in whole blood in GTEx data [97], with the minor allele associated with lower *TDRD9* expression. Although evidence from this study points to *PPP1R13B* in the airway epithelium, we can't exclude the possibility that *TDRD9* or other genes could contribute to the locus' influence on lung function via other tissues.

The FEV$_1$ association signal at the *TDRD9* locus included many variants in high LD across a 200 kb region that could be independently contributing to function. Some of the variants lie in different long-range enhancers [59]. It is also possible that one or more correlated variants were not included because they failed quality control standards. In addition, due to the limited sample size of the WGS cohort, we excluded rare variants (MAF<0.01) from consideration,

which could contribute to the signal at this locus. Additional functional studies are needed to identify the causal variant(s) and full mechanism of action.

The correlations of rs10220464, $FEV_1$, and smoking exposure with cg03306306 methylation in NECs were absent in PBMCs. Although global DNA methylation patterns between tissues are highly correlated [98], tissue-specific differentially methylated regions are more likely to be functional, particularly if they are positively correlated with gene expression [99]. The *TDRD9* locus has not been identified in epigenome-wide association studies of lung function [44,100–104], but these measured DNA methylation from blood, which may be an insufficient proxy for methylation in the lungs [105]. Indeed, previous studies have found that DNA methylation profiles in NECs are significantly more predictive of pediatric asthma than those in PBMCs [106,107]. Furthermore, epigenetic biomarkers can change with age. For example, epigenetic markers for lung function in adults do not replicate in children [101].

We tested for interactions between smoking exposure and rs10220464 genotype effects on cg03306306 and on $FEV_1$ and between smoking exposure and cg03306306 methylation effects on $FEV_1$. We did not detect any significant interactions, but our analyses in that regard could have been underpowered given our observed effects and sample sizes [36]. Furthermore, because this study was limited to children living in low-income urban neighborhoods, environmental risk factors are likely to be more prevalent than in the general population [55–57]. Additionally, such exposures are not necessarily ubiquitous across all the different neighborhoods and communities represented in this sample, and although environmental tobacco smoke exposure was examined and the socioeconomic range represented in this study is relatively narrow, there could be relevant environmental factors that were not considered.

To infer causality, Mendelian randomization and mediation analyses rely on assumptions that are often difficult to empirically verify. For the Mendelian randomization analysis, we identified instrumental variants associated with the intermediate cg03306306 that were not independently associated with the outcome, $FEV_1$. However, because these variants were selected from the same dataset that the outcome testing was performed in, they were susceptible to bias from winner's curse [108]. To mitigate the potential impact from this effect and from weak instruments, we performed a secondary analysis in which we combined the instrumental variants into a single, unweighted score. For the mediation analysis, unmeasured confounding can invalidate direct and indirect effect estimates [109]. To protect against such bias, we systematically tested for confounding associations with additional environmental measures available in APIC and URECA (Materials and methods). Nonetheless, there may still exist unknown confounding factors that were not measured. Ultimately the results of the Mendelian randomization and mediation analyses indicate that methylation at cg03306306 in NECs mediated the rs10220464 genotype effect on $FEV_1$, but there was residual correlation between rs10220464 and $FEV_1$, signifying that the genotype effect was only partially mediated by cg03306306.

Another limitation of our study was the relatively small size for a GWAS. This likely contributed to the lack of statistically significant replication for previously identified lung function loci [23], considering that the observed effects were correlated with results of prior GWAS. However, the APIC and URECA cohorts represent understudied, high-risk, pediatric populations that likely harbor distinct genetic and environmental risk factors compared to older, primarily European ancestry cohorts included in previous GWAS of lung function [14–20,23]. The findings of this study have yet to be replicated in an independent cohort, and should therefore be considered preliminary; however, it is possible that these associations would differ in populations with dissimilar ancestry, age, exposures, and/or asthma risk.

There are additional caveats to consider when interpreting our findings. First, this study integrated data from two cohorts with different recruitment criteria, asthma definitions, and

ancestral compositions. Furthermore, most of the analyses beyond the GWAS were limited to subsets of the URECA participants. However, we did not observe significant genetic effect heterogeneity for rs10220464 by study, asthma status, or ancestry. To control for potential population stratification, we used the first ten PCs of ancestry to adjust lung function values and then included the ancestry PCs as fixed effects in the GWAS models (Materials and methods). The linear mixed models also included a genetic relatedness matrix as a random effect to account for residual population structure. Because children with asthma have lower lung function overall (Table 1) and their lung function may be more affected by environmental exposures [110–112], we adjusted for asthma status in the GWAS, as in previous GWAS [113–116]. The likelihood of discovering lung function variants with consistent effects in asthmatics and non-asthmatics was thereby increased, although genetic determinants of lung function may differ by asthma status [117]. Furthermore, adjusting for disease status could potentially introduce collider bias [118]. The significant genotype effect at the *TDRD9* locus, however, remained the only genome-wide-significant association when asthma was excluded as a covariate, and adjustment for asthma did not substantively alter the mediation results. Second, some of the analyses used data collected at different timepoints. For example, most of the urine cotinine and spirometry measures were collected at age 10, but the samples used for the NEC DNA methylation and RNA-seq analyses were collected at age 11. Because DNA methylation and gene expression can change over time [40,119–121], their values at age 11 may not be fully representative of exposures at age 10. Finally, the promoter-capture Hi-C data were from lower airway (bronchial) epithelial cells, whereas the DNA methylation and RNA-seq data were generated from upper airway (nasal) epithelial cells. Although there are transcriptomic differences between epithelial cells from each compartment, their respective profiles are highly correlated [122–126], and the use of NECs as a proxy for the lower airway epithelium has been validated for both gene expression and epigenetic studies [124–127].

Our study identified a novel avenue through which genetic risk and environmental exposures could affect the airways of children raised in low-income urban neighborhoods. Further research into this pathway may yield mechanistic insights into the early development of impaired lung function, perhaps leading to interventions that can help reduce the high incidence and morbidity of chronic respiratory diseases in socioeconomically disadvantaged children.

## Materials and methods

### Ethics statement

The institutional review boards (IRBs) from all participating sites of the URECA (ClinicalTrials.gov Identifier: NCT00114881) and APIC (ClinicalTrials.gov Identifier: NCT01383941) studies gave initial ethical approval for this work. These include IRBs from the following institutions: National Jewish Health, Denver, CO (APIC); Children's National Medical Center, Washington, DC (APIC); Children's Memorial Hospital, Chicago, IL (APIC); Johns Hopkins University, Baltimore, MD (APIC & URECA); Boston University School of Medicine, Boston, MA (APIC); Henry Ford Health Center, Detroit, MI (APIC); Columbia University Medical Center, New York, NY (APIC & URECA); Cincinnati Children's Hospital, Cincinnati, OH (APIC); University of Texas Southwestern Medical School, Dallas, TX (APIC); Boston Medical Center, Boston, MA (URECA); Saint Louis Children's Hospital, Saint Louis, MO (URECA). In 2014, ethical oversight for these studies transitioned to a single, central IRB managed by WGC IRB (formerly Western IRB), whereupon WGC IRB gave ethical approval for this work [128]. Written informed consent was obtained from legal guardians of all participating children, who also assented.

## Study population and phenotypes

We analyzed samples and phenotypes from two National Institutes of Allergy and Infectious Diseases (NIAID)-funded asthma studies conducted by the Inner-City Asthma Consortium (ICAC) [129]: the Asthma Phenotypes in the Inner City (APIC) study [55,56] and the Urban Environment and Childhood Asthma (URECA) birth cohort study [57]. The APIC study was a 1-year, prospective, epidemiological investigation of children and adolescents with asthma (ages 6–17) living in low-income areas ($\geq$20% of residents below poverty level) in nine U.S. cities (Baltimore, MD; Boston, MA; Chicago, IL; Cincinnati, OH; Dallas, TX; Denver, CO; Detroit, MI; New York, NY; Washington, DC). The APIC participants were required to have a diagnosis of asthma by a physician and to have had at least two episodes requiring bronchodilator administration within the past year [55]. The URECA study enrolled pregnant women living in low-income areas of four U.S. cities (Baltimore, MD; Boston, MA; New York, NY; St. Louis, MO) who reported that either or both parents of the index pregnancy had a history of asthma or allergic diseases [57]. This prospective, longitudinal study followed each child through adolescence, periodically collecting samples and clinical and environmental exposure data.

Lung function was assessed using spirometry. Lung function measures used in this study for APIC participants were taken at the study entry visit (V0). For URECA, measurements from age 10 were used when available; otherwise, the most recent measurement after age 5 was used (S6 Table). Asthma status was assigned according to study-specific criteria. For APIC, asthma was defined by a doctor's diagnosis of asthma and short-acting beta-agonist use in the year prior [55]. For URECA, asthma status was determined either by doctor diagnosis, lung function reversibility, or symptom recurrence [130]. The 2012 Global Lung Initiative reference equations [131] were applied to generate percent predicted estimates for $FEV_1$ and Z-scores for $FEV_1$/FVC ratio. Urine cotinine levels were measured using NicAlert immunochromatographic assays, which report results on a scale of 0–6 according to different cotinine concentration ranges [132]. For URECA, urine cotinine results were available at age 10 for most participants (n = 391); otherwise, assays from age 8 (n = 29) or age 7 (n = 2) were used. This study utilized DNA methylation and RNA-seq data generated for other URECA studies; therefore, the number of samples included in each analysis varied and was limited by data availability (S7 Table, S12 Fig).

## Whole-genome sequencing and data processing

DNA was extracted from peripheral blood (APIC, URECA) or cord blood (URECA) and quantified using an Invitrogen Qubit 3 Fluorometer. DNA quality was assessed using the Thermo Scientific NanoDrop One spectrophotometer and confirmed using an Agilent TapeStation system. DNA was processed in batches of 60 using the Illumina Nextera DNA Flex library prep kit with unique dual adaptors. Each set of 60 libraries was sequenced over two NovaSEQ S4 flowcells. Whole-genome sequencing was performed by the University of Chicago Genomics Facility using the Illumina NovaSEQ6000, which generated 150 bp paired-end reads. Sequencing data processing followed the Broad Institute's Genome Analysis Toolkit (GATK) best practices for germline short variant discovery, as implemented in the harmonized pipeline used by the New York Genome Center for TOPMed [133,134]. Reads were aligned to the GRCh38 human reference genome (including alternate loci and decoy contigs) using BWA-MEM (Burrows-Wheeler Aligner; v0.7.17). Aligned reads further underwent duplicate removal (Picard MarkDuplicates; v2.8.1) and base quality score recalibration (GATK BaseRecalibrator; v3.8) against known sites (dbSNP138, known InDels, and Mills and 1KG gold standard InDels) provided in the GATK resource bundle. Read alignment metrics were

calculated using Picard CollectWgsMetrics (v2.8.1) for all aligned reads and for aligned reads with base quality and mapping quality $\geq$ 20. DNA contamination levels were estimated using VerifyBamID2 (v1.0.6) [135]. Samples with estimated DNA contamination >0.05 were removed from consideration. Samples with poor coverage (<50% of the genome with $\geq$20x depth) were also removed from further consideration. To identify potential sample swaps, WGS samples were validated using independent genotyping arrays.

### QC array for sample validation

To identify potential WGS sample swaps, we independently genotyped the APIC and URECA participants using the Illumina QC Array-24 BeadChip. SNPs were tested for Hardy-Weinberg Equilibrium (HWE) within each self-identified ancestry group using the chi-square test and removed if they deviated from HWE (Bonferroni-adjusted p<0.05) within at least one ancestry. SNPs with call rates <0.98 were also removed. Samples with total variant call rates <0.95 were not used. Array data with incorrect or indeterminate sex according to X-chromosome heterozygosity rates (Plink v1.90) were also not used [136]. For fourteen of the sequenced URECA samples, we used results from the Illumina Infinium CoreExome+Custom array for sample validation, which were generated and controlled for quality as described by McKennan and colleagues [137]. WGS and array genotypes were tested for concordance using VerifyBamID (v1.1.3) [138]. WGS samples that were not validated with array data were not included in genetic analyses (n = 2).

### Variant calling and quality control

Variant calls were generated using GATK HaplotypeCaller (v4.1.3.0), accounting for contamination estimates, for single nucleotide variants and short insertions, deletions, and substitutions. Sample genotypes were joined using GATK GenomicsDBImport and GenotypeGVCFs over the genomic intervals defined in the GATK WGS calling region interval list provided in the GATK resource bundle. Genotypes with read depth (DP) <10 or quality scores (GQ) <20 were set as missing. Sites with $\geq$0.1 missingness were then removed from consideration. Variants with minor allele frequencies >0.05 were tested for accordance with HWE, accounting for population structure [139]. Sites with common variants that deviated from structural HWE (P<1x10$^{-6}$) were removed from consideration. Sites with quality by depth ratios (QD) <4 or >34 were also removed, as we observed declines in variant transition/transversion (TS/TV) ratios beyond these bounds (S13 Fig). Variant site quality was further evaluated using machine-learning-based Variant Quality Score Recalibration (VQSR). First, SNPs were modeled using GATK VariantRecalibrator (v4.1.3.0) with Hapmap 3 and with Omni 2.5M SNP chip array as truth resources, 1000G as a training resource, and dbSNP138 as a known sites resource. InDels were likewise trained with the Mills and 1KG gold standard InDels dataset as a truth resource and dbSNP138 as a known sites resource. SNPs and InDels with resultant predicted true positive probabilities below 0.997 and 0.990, respectively, were removed from consideration. Variant call accuracy was assessed by comparing call concordance between three replicate sequencing samples using VCFtools (v0.1.14) vcf-compare [140]. Variant call file manipulation was conducted using BCFtools (v1.10.2) [141].

### Ancestry estimation

Ancestry principal components (PCs) were calculated on the intersect of high quality single-nucleotide variants (SNVs) genotyped in the WGS data and several reference panels from the 1000 Genomes Project (1KG; n = 156) [142] and the Human Genome Diversity Project (HGDP; n = 52) [143]. Native American reference samples consisted of 52 samples from the

HGDP with <5% non-native ancestry, according to an analysis of roughly 2 million markers using the program ADMIXTURE (v1.3.0) [144]. These samples were filtered for site quality (missingness 5%; ExcHet<60; VQSLOD≥8.3929), genotype quality (GQ≥20) and depth (DP≥10), MAF >0.02, and HWE (p>0.001) [143]. European, West African, and East Asian reference samples were randomly selected from CEU (n = 52), YRI (n = 52), JPT (n = 26), and CHB (n = 26) samples in the phase 3 1KG reference panel [142]. The combined genotypes were pruned for linkage disequilibrium (LD) ≤0.1 within 1Mb intervals. Ancestry PCs were calculated, accounting for subject relatedness, using PC-Air [145] and PC-Relate [146]. Initial kinship estimates were produced using KING [147]. Kinship and PCs were iteratively estimated using PC-Relate and PC-Air, respectively, until estimates for the top 5 PCs stabilized (n = 3). Reference population admixture estimates were estimated for each WGS sample with ADMIXTURE (v1.3.0), using the 1KG and HGDP reference samples for supervised analysis [144]. Because sample relatedness can lead to biased admixture estimates [145,148], admixture was estimated for each WGS sample separately.

### Quantitative trait association testing

Quantitative traits were adjusted for covariates and normalized using a two-stage approach [149,150]. First, each trait was regressed on age, sex, asthma status, and the first 10 PCs of ancestry. The residuals were then rank-normalized using an inverse normal transformation. In the second stage, the normalized residuals were considered outcome variables in the GWAS, adjusting for the same covariates as in the first stage. Genome-wide association testing was performed for all high-quality common variant calls (MAF≥0.01) using a linear mixed model, as implemented in GEMMA [58], with subject relatedness included as a random effect. Individuals who were not evaluated for asthma at ages 7 or 10 (n = 127) were excluded from trait association testing. The threshold we applied for genome-wide significance was $P \leq 2.5 \times 10^{-8}$, based on a $5 \times 10^{-8}$ GWAS threshold and further accounting for two tests. To identify potential collider bias introduced by adjusting for asthma status, we repeated the GWAS without accounting for asthma status in either covariate-adjustment stage.

Fine-mapping analysis was conducted using SuSiE (SusieR R package v0.12.27) [151]. SuSiE applies a form of Bayesian variable selection in regression using iterative Bayesian stepwise selection to identify "credible sets" of variables. Each credible set has a 95% probability of containing at least one causal effect SNP. Prior to running SuSiE, we regressed asthma, age, sex, and ancestry PCs 1–10 from the genotype matrix and outcome vector (the normalized $FEV_1$ residuals).

To explore whether there was lead-SNP effect heterogeneity by ancestry, study, or asthma status, we performed additional single-SNP quantitative trait association tests within several different sub-cohorts and introduced interaction effects into our models. For ancestry, we performed separate association tests in each of the non-Hispanic Black, Hispanic, and white populations, according to self-identified race/ethnicity. We then tested for genotype-by-ancestry interaction effects across APIC and URECA by using admixture proportions as covariates in our models, in lieu of ancestry PCs, and including an interaction term with the lead SNP for each continental ancestry group in turn. We tested these interaction effects using the—gxe argument in GEMMA in four separate models (one for each ancestry). To determine whether there was effect heterogeneity by study (APIC vs. URECA), we performed separate association tests in each study and also tested the association across APIC and URECA with the addition of a study covariate and a genotype-by-study interaction term. For asthma status, we performed separate association tests in the asthmatics and non-asthmatics and tested a genotype interaction term with asthma status.

## DNA methylation analysis

DNA from NECs was collected at age 11 from 287 URECA participants and assessed for genome-wide methylation patterns using the Illumina Infinium Human Methylation EPIC Beadchip. DNA methylation levels from PBMCs at age 7 in URECA were collected and processed as previously described [66]. MeQTL analysis was performed using Matrix eQTL [152]. NEC DNA methylation levels were adjusted globally for sex, array, plate, collection site, DNA concentration, percent ciliated epithelial cells, percent squamous cells, and ancestry PCs 1–3. Principal components analysis was then performed on the residual methylation levels, and the first three PCs were included as covariates in the meQTL association tests. Additional methylation PCs were not included in association tests, as they were significantly correlated with asthma phenotypes. Associations with FDR-adjusted $P<0.05$ were considered significant. MeQTL analysis with the PBMC data included sex, collection site, plate, ancestry PCs 1–3, and eight latent factors [153] (protecting for $FEV_1$ at age 7) as covariates.

To test CpG site methylation associations with lung function in NECs, we performed linear regressions on the most recent $FEV_1$ measures, with age, sex, ancestry PCs 1–3, and methylation PCs 1–3 as covariates. For the PBMC analysis, we set $FEV_1$ at age 7 as the dependent variable, with sex, collection site, plate, ancestry PCs 1–3, and latent factors included as covariates.

For association testing with smoking exposures, we ran linear regressions for DNA methylation and lung function in NECs and PBMCs, as described above, with the addition of cotinine concentrations as a predictor. We further tested for smoking-by-genotype interaction effects on DNA methylation and lung function using these models by adding an interaction term (cotinine concentration: rs10220464 genotype). Proportions of explained variance were calculated by squaring partial correlation coefficients of regression model predictors [154]. One sample from one sibling pair was removed from all methylation analyses to prevent confounding due to relatedness.

## Mendelian randomization and mediation analysis

To assess the causal effects of DNA methylation on lung function, we performed one-sample Mendelian randomization analysis. We applied a 2SLS regression to URECA samples with WGS and DNA methylation data (n = 285) using ivreg [155]. DNA methylation levels in NECs at the cg03306306 CpG site were first adjusted for methylation PCs 1–3 and used as the endogenous, exposure variable. The adjusted and normalized $FEV_1$ values from the GWAS were set as the dependent outcome variable. Urine cotinine levels were included as an exogenous covariate (included in both stages). The instrumental variables were chosen from a set of candidate SNPs that were at least nominally associated with cg03306306 methylation with $p<0.15$. Clustering of pairwise linkage disequilibrium values between these SNPs revealed six distinct haplotypes (S14 Fig). To ensure instrument exogeneity, each candidate SNP was tested for association with $FEV_1$ after conditioning on cg03306306 methylation and urine cotinine, and SNPs associated with $p<0.05$ were removed from consideration. Of the remaining candidate SNPs, one was chosen from each haplotype, resulting in an instrument composed of 4 SNPs (rs11160777, rs137961671, rs7143936, rs11160776). Instrument relevance was validated using the F test, endogeneity using the Wu-Hausman test, and instrument exogeneity using the Sargon test. We tested two 2SLS models: one where the instrumental variables were included as individual predictors, and another featuring an unweighted allele score of the four instrumental variants to reduce potential bias from weak instruments and/or winner's curse [156,157].

Mediation analysis was conducted with ROBMED [158]. The adjusted and normalized $FEV_1$ residuals were set as the dependent variable, adjusted cg03306306 methylation as the mediator,

and rs10220464 as the independent variable. Age at $FEV_1$ measurement, sex, asthma status, ancestry PCs 1–3, and urine cotinine levels were included as covariates. We also performed a secondary mediation analysis without adjusting for asthma status. To identify additional, potential confounders that could invalidate our mediation model, we systematically tested for associations with 2 socioeconomic variables and 11 environmental exposures available in APIC and URECA (S8 Table, S15 Fig). For each environmental exposure, we tested whether the variable was associated with the mediator (cg03306306) and whether the variable was associated with the outcome ($FEV_1$) conditional on the mediator. To ensure no exposure-mediator interactions, we repeated the cg03306306 association test with $FEV_1$ with rs10220464 included as a predictor with a rs10220464: cg03306306 interaction term. The indirect effect of rs10220464 on $FEV_1$ via cg03306306 methylation was estimated using 100,000 bootstrap resamples.

### Gene expression analysis

We analyzed gene expression in NECs and PBMCs from the URECA birth cohort using RNA-seq. The NEC data were derived from 323 children (155 females, 168 males) at age 11 years at the time of sample collection, and the PBMC data were derived from 130 (53 females, 77 males) PBMC children aged 2 years at the time of collection. Sequencing reads were mapped and quantified using STAR (v2.6.1) [159] and samples underwent trimmed means of M-value (TMM) normalization and voom transformation [160]. Genes with <1 count per million mapped reads (CPM) were removed from analysis. For eQTL association testing in NECs we corrected for sex, the first three ancestry PCs, collection site, epithelial cell proportion, sequencing batch, and 14 latent factors [153] using limma [161]. In PBMCs, we corrected for sex, the first three ancestry PCs, collection site, and 19 latent factors.

### Chromatin interaction analysis

Chromatin interactions were assessed using promoter capture Hi-C [162,163] in ex vivo human BECs from 8 adult lung donors, including 4 with asthma. The data were processed and analyzed as previously described [72,164]. Chromosomal interactions were evaluated using the CHiCAGO algorithm [165]. Interactions with CHiCAGO scores ≥5 were considered significant [165]. Genetic variants within 1 kb of a given interacting fragment were considered part of the chromatin loop. Genes that were not expressed in NECs were not included in the analysis.

### Supporting information

**S1 Fig. Whole-genome sequencing depth and coverage. A)** Histogram of 1,035 whole-genome sequencing (WGS) samples from APIC and URECA by mean depth of coverage. **B)** Histogram of WGS samples based on proportion of genome covered at 20x, 25x, and 30x depth. APIC, Asthma Phenotypes in the Inner City study; URECA, Urban Environment and Childhood Asthma study.
(PDF)

**S2 Fig. Distribution of lung function measures by study. A)** Distribution of $FEV_1$ (% predicted) in APIC and URECA. **B)** Distribution of $FEV_1$/FVC in APIC and URECA. APIC, Asthma Phenotypes in the Inner City study; URECA, Urban Environment and Childhood Asthma study. $FEV_1$, forced expiratory volume in one second; FVC, forced vital capacity.
(PDF)

**S3 Fig. P-value distributions of GWAS results.** Quantile-quantile plots of the GWAS results with corresponding genomic control factors (lambda) are shown for A) $FEV_1$ (% predicted)

and B) FEV$_1$/FVC. FEV$_1$, forced expiratory volume in one second; FVC, forced vital capacity.
(PDF)

**S4 Fig. Fine-mapping results for FEV$_1$ (% predicted) at the *TDRD9* locus.** The X-axis shows the chromosome position on chromosome 14. The Y-axis is the posterior inclusion probability (PIP). Variants highlighted in red represent a credible set, in which there is a 95% probability that at least one of the variants is causal. FEV$_1$, forced expiratory volume in one second.
(PDF)

**S5 Fig. Genome-wide association results without adjustment for asthma.** GWAS Manhattan plots for **A**) FEV$_1$ and **B**) FEV$_1$/FVC ratio, without adjustment for asthma status. The horizontal red line indicates genome-wide significance (p $\leq$ 2.5x10$^{-8}$). The dotted horizontal blue line indicates p = 1x10$^{-5}$. Variants colored in grey are the GWAS results with asthma adjustment. FEV$_1$, forced expiratory volume in one second; FEV$_1$/FVC, ratio of FEV$_1$ to forced vital capacity.
(PDF)

**S6 Fig. Replication of FEV$_1$ GWAS SNPs.** Association statistics for previously identified FEV$_1$ GWAS SNPs [23]. 64 out of 70 previously identified SNPs were genotyped in APIC & URECA. GWAS, genome-wide association study; SNP, single nucleotide polymorphism; APIC, Asthma Phenotypes in the Inner City study; URECA, Urban Environment and Childhood Asthma study. FEV$_1$, forced expiratory volume in one second.
(PDF)

**S7 Fig. Replication of FEV$_1$/FVC GWAS SNPs.** Association statistics for previously identified FEV$_1$/FVC GWAS SNPs [23]. 112 out of 117 previously identified SNPs were genotyped in APIC & URECA. GWAS, genome-wide association study; SNP, single nucleotide polymorphism; APIC, Asthma Phenotypes in the Inner City study; URECA, Urban Environment and Childhood Asthma study. FEV$_1$, forced expiratory volume in one second; FVC, forced vital capacity.
(PDF)

**S8 Fig. NicAlert Results by Study.** Distribution of urine cotinine levels, as measured using NicAlert immunochromatographic assays, which report results on a scale of 0–6 according to the labeled concentration ranges. Proportions were calculated relative to the number of samples with available NicAlert results. APIC, Asthma Phenotypes in the Inner City study; URECA, Urban Environment and Childhood Asthma study.
(PDF)

**S9 Fig. DNA methylation at cg03306306 by smoking exposure.** DNA methylation levels at cg03306306 are shown by rs10220464 genotype in URECA participants with low and high smoking exposures in (**A**) NECs at age 11 and (**B**) PBMCs at age 7. FEV$_1$ (% predicted) are also shown by cg03306306 DNA methylation levels in URECA participants with low and high smoking exposures in (**C**) NECs at age 11 and (**D**) PBMCs at age 7. NECs, nasal epithelial cells; PBMCs, peripheral blood mononuclear cells; FEV$_1$, forced expiratory volume in one second; URECA, Urban Environment and Childhood Asthma study.
(PDF)

**S10 Fig. Genotype associations with FEV$_1$ by smoking exposure.** FEV$_1$ (% predicted) are shown by rs10220464 genotype in APIC & URECA participants with low and high smoking exposures according to urine cotinine levels. FEV$_1$, forced expiratory volume in one second; APIC, Asthma Phenotypes in the Inner City study; URECA, Urban Environment and

Childhood Asthma study.
(PDF)

**S11 Fig. *PPP1R13B* expression in NECs vs. smoking exposure, FEV$_1$.** *PPP1R13B* expression in NECs at age 11 was not associated with smoking exposure at age 10 (**A**) nor with FEV$_1$ (% predicted) at age 10 (**B**) in URECA. NECs, nasal epithelial cells; FEV$_1$, forced expiratory volume in one second; URECA, Urban Environment and Childhood Asthma.
(PDF)

**S12 Fig. Data availability across APIC and URECA.** Data availability for measures used in this study are shown for all sequenced samples. Each row represents a pattern of available and missing data, with green squares indicating available data and grey squares indicating missing data. Total counts of available data points for each variable are listed across the top of the figure. Total counts for each data availability pattern are listed along the right.
(PDF)

**S13 Fig. Transitions/transversions vs. quality/depth in WGS variant calls.** The transition/transversion ratio (TS/TV) is plotted against the variant call quality/depth metric (QD) across all WGS SNP calls in APIC & URECA. Sites with QD less than 4 or greater than 34 were removed from consideration in this study. SNPs, single nucleotide polymorphisms; WGS, whole-genome sequencing; APIC, Asthma Phenotypes in the Inner City study; URECA, Urban Environment and Childhood Asthma study.
(PDF)

**S14 Fig. Intercorrelation of Mendelian randomization candidate instrument SNPs in URECA.** Instrumental variables were chosen from a set of candidate SNPs that were at least nominally associated with cg03306306 methylation with p<0.15. The correlation values between these SNPs are shown, clustered using Ward's method. The four SNPs used for the instrument are highlighted. URECA, Urban Environment and Childhood Asthma.
(PDF)

**S15 Fig. Intercorrelation of phenotypes and environmental variables in APIC & URECA.** The correlations are shown between FEV$_1$ (% predicted), smoking exposure (NicAlert), the primary the lead FEV$_1$ SNP rs10220464, DNA methylation at cg03306306, 11 environmental exposures, and 2 socioeconomic indicators, clustered using Ward's method. APIC, Asthma Phenotypes in the Inner City study; exp., exposure; URECA, Urban Environment and Childhood Asthma.
(PDF)

**S1 Table. Post-QC sequencing call concordance between replicates.** Variant call concordance between three pairs of replicate samples, by variant type and cohort allele frequency. SNPs, single nucleotide polymorphisms; MAF, minor allele frequency; InDels, insertions and deletions.
(PDF)

**S2 Table. FEV$_1$-associated variants in chr14q32.33.** All variants in chr14q32.33 associated with FEV$_1$ (% predicted) with p<1x10$^{-5}$ (n = 82) in GWAS of 896 participants from APIC & URECA. N, number of genotyped individuals. MAF, minor allele frequency; 95% CI, 95% confidence interval; SE, standard error; P, P-value (Wald); FEV$_1$, forced expiratory volume in one second; APIC, Asthma Phenotypes in the Inner City study; URECA, Urban Environment and Childhood Asthma study.
(PDF)

**S3 Table. MeQTL analysis results and associations with FEV$_1$.** All CpG sites where DNA methylation levels in NECs at age 11 in URECA were associated with rs10220464 at FDR<0.05 are shown with their corresponding associations with FEV$_1$. The FDR-adjusted P-values (FDR Q) correspond to a 5% false-discovery rate. FDR, false discovery rate; 95% CI, 95% confidence interval; FEV$_1$, forced expiratory volume in one second; URECA, Urban Environment and Childhood Asthma study.
(PDF)

**S4 Table. rs10220464 eQTL analysis results.** Results of eQTL analyses in NECs and PBMCs with rs10220464 for all genes within 1 Mb in URECA. Gene expression was measured in counts per million mapped reads. The FDR-adjusted P-values (FDR Q) correspond to a 5% false-discovery rate. FDR, false discovery rate; 95% CI, 95% confidence interval; NECs, nasal epithelial cells; PBMCs, peripheral blood mononuclear cells; URECA, Urban Environment and Childhood Asthma study.
(PDF)

**S5 Table. Chromatin interactions with FEV$_1$-associated SNPs.** Bait and target fragments refer to mapped Hi-C restriction fragments on chr14 (hg38) for gene promoters and putative enhancers, respectively. FEV$_1$ SNPs refer to number of FEV$_1$-associated variants (p<1x10$^{-5}$) within 1kb of target fragment. SNPs, single nucleotide polymorphisms; FEV$_1$, forced expiratory volume in one second.
(PDF)

**S6 Table. Age at used lung function measure in URECA.** URECA, Urban Environment and Childhood Asthma study; FEV$_1$, forced expiratory volume in one second; FVC, forced vital capacity.
(PDF)

**S7 Table. Study samples.** APIC, Asthma Phenotypes in the Inner City study; URECA, Urban Environment and Childhood Asthma study; WGS, whole-genome sequencing; NECs, nasal epithelial cells; PBMCs, peripheral blood mononuclear cells.
(PDF)

**S8 Table. Additional phenotypic, socioeconomic, and environmental data.** Additional variables examined for potential confounding in mediation analyses for APIC & URECA. APIC, Asthma Phenotypes in the Inner City study; URECA, Urban Environment and Childhood Asthma study.
(PDF)

## Acknowledgments

We are grateful to all participants and their families who took part in these studies. We would like to extend special thanks to Pieter Faber and the University of Chicago Genomics Facility, as well as to Petra LeBeau and Rebecca Z. Krouse, formerly of Rho Inc., for their respective contributions.

## Author Contributions

**Conceptualization:** Matthew Dapas, William W. Busse, James E. Gern, Daniel J. Jackson, Carole Ober.

**Data curation:** Matthew Dapas, Emma E. Thompson, William Wentworth-Sheilds, Cynthia M. Visness, Agustin Calatroni.

**Formal analysis:** Matthew Dapas, Emma E. Thompson, Selene Clay, Agustin Calatroni, Joanne E. Sordillo, Matthew C. Altman.

**Funding acquisition:** Matthew Dapas, William W. Busse, James E. Gern, Daniel J. Jackson, Carole Ober.

**Investigation:** Matthew Dapas, Robert A. Wood, Melanie Makhija, Gurjit K. Khurana Hershey, Michael G. Sherenian, Rebecca S. Gruchalla, Michelle A. Gill, Andrew H. Liu, Haejin Kim, Meyer Kattan, Deepa Rastogi, George T. O'Connor.

**Methodology:** Matthew Dapas, Dan Nicolae.

**Project administration:** Matthew Dapas, William Wentworth-Sheilds, Cynthia M. Visness, Leonard B. Bacharier, William W. Busse, Patrice M. Becker, George T. O'Connor, James E. Gern, Daniel J. Jackson, Carole Ober.

**Resources:** Carole Ober.

**Software:** Matthew Dapas.

**Supervision:** Cynthia M. Visness, Leonard B. Bacharier, William W. Busse, Patrice M. Becker, Dan Nicolae, George T. O'Connor, James E. Gern, Daniel J. Jackson, Carole Ober.

**Validation:** Matthew Dapas, Joanne E. Sordillo, Diane R. Gold.

**Visualization:** Matthew Dapas.

**Writing – original draft:** Matthew Dapas.

**Writing – review & editing:** Matthew Dapas, George T. O'Connor, James E. Gern, Daniel J. Jackson, Carole Ober.

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
