## [Decision Letter · Decision Letter 0]

26 Jul 2022

Dear Dr Dapas,

Thank you very much for submitting your Research Article entitled 'Multi-omic association study implicates PPP1R13B in DNA methylation-mediated genotype and smoking exposure effects on decreased lung function in urban children' to PLOS Genetics.

The manuscript was fully evaluated at the editorial level and by independent peer reviewers. The reviewers appreciated the attention to an important problem, but raised some substantial concerns about the current manuscript. Based on the reviews, we will not be able to accept this version of the manuscript, but we would be willing to review a much-revised version. We cannot, of course, promise publication at that time.

Should you decide to revise the manuscript for further consideration here, your revisions should address the specific points made by each reviewer. We will also require a detailed list of your responses to the review comments and a description of the changes you have made in the manuscript. Additionally,  concerns raised by Reviewer 2 regarding data accessibility should be clearly addressed. 

If you decide to revise the manuscript for further consideration at PLOS Genetics, please aim to resubmit within the next 60 days, unless it will take extra time to address the concerns of the reviewers, in which case we would appreciate an expected resubmission date by email to plosgenetics@plos.org.

[LINK]

We are sorry that we cannot be more positive about your manuscript at this stage. Please do not hesitate to contact us if you have any concerns or questions.

Yours sincerely,

Wei Pan

Guest Editor

PLOS Genetics

Hua Tang

Section Editor: Human Variation

PLOS Genetics

Reviewer's Responses to Questions

**Comments to the Authors:**

Reviewer #1: Authors looked into an understudied multiethnic pediatric population of sample size of 1035 and discovered one significant locus on 14q32.22 from GWAS of reduced lung function. Authors then moved on to study its functionality through locating its corresponding DNA methylation (which was linked to smoking), followed by additional analyses looking into the gene expression and promoter-enhancer interaction. The manuscript is overall clearly written.

Detailed comments:

1. It would be helpful to mention the sample size and the study population of Shrine’s meta-analysis study (line 171)

2. A fine mapping analysis would be appropriate here to narrow down the causal SNP in the region given the multi-ancestral sample, as authors also pointed out in line 342. The conclusions of the manuscript are mainly based on this SNP rs10220464 which has the highest significance. From Fig 3, there are really quite a few SNPs with similar p-value and highly correlated with this SNP. Is there any public available GWAS dataset for European children?

3. Rs10220464 was reported to associated with the methylation at TDRD9, TDRD9 was associated with maternal smoking during pregnancy. Is the SNP reported to be associated with smoking?

4. Report the variance explained in stage 1 of the MR model (line 232)

5. Plotting a graph demonstrated the relationship among leading SNP, CpG site, gene expression, smoking and lung function would be helpful.

6. Report the genetic inflation factor of GWAS study or provide other evidence of the control of population stratification. Are 3 PCs enough to control for population stratification in methylation association analysis, gene expression analysis and mediation analysis for the multiethnic population?

7. One strength of the manuscript is that the gene expression and methylation were both measured from the same pediatric population.

Reviewer #2: General comments: Dapas and coauthors presented an integrative study of lung function in two multi-ethnic cohorts of children living in underprivileged, urban communities by interrogating GWAS, DNA methylation, RNA sequencing, and HiC data. Given that most GWAS of lung functions have focused on adult populations of primarily European ancestry, this study is a timely contribution to the literature and provides unique insights into the genetic, genomic and environmental factors underlying lung function in multi-ethnic, underprivileged children. The manuscript is overall well written and I enjoyed reading it. It is a considerable strength to interrogate DNA methylation and RNAseq in lung function-relevant tissues (upper airway (nasal) epithelial cells (NECs) and lower airway (bronchial) epithelial cells (BECs)), as well as PBMC and blood, which contributes to the novel insights provided by this study. I have some specific comments about the analysis methods and results interpretation as follows.

1. The role of asthma: based on Table 1, one of the two cohorts (APIC) only consisted of children with asthma, while the other cohort is also substantially over-represented by kids with asthma. This reviewer is thus wondering how much the identified signals are attributed to asthma. Also the manuscript seemed inconsistent regarding how asthma was treated in the analyses. For example, it states “we adjusted for asthma status in the GWAS” in line 367, while it states “Individuals without a confirmed asthma status (n=127) were excluded from trait association testing” in line 497. Please clarify.

2. In addition to Manhattan plots (Fig 2), the authors should present QQ plots as well as genomic control lamba.

3. For the mediation analysis, could the authors also present proportion (percentage) of total effect mediated by the CpG site methylation?

4. The lead SNP rs10220464’s MAFs differ across ethnicities according to dbSNP (lower in Blacks and higher Hispanics and Whites). Could the authors report the MAFs for this SNP in the current cohorts by ethnic groups, in particular, in Blacks and Hispanics. Was there indication of genetic effect heterogeneity across Black/Hispanic, asthma/non-asthma, and between the two cohorts?

5. line 186: please clarify n=82 was the sample size or # of GWAS loci.

6. lines starting from 228 and lines starting from 525 regarding MR analysis: it seems that the four SNPs/IVs used in the MR analysis were derived from the dataset itself based on association testing, which may suffer from the Winner’s curse. Also were the four selected SNPs used as four IVs or a single IV by constructing a weighted/unweighted genetic score?

7. data availability: WGS data phs002921.v1.p1 cannot be found on dbGaP. There is no information about where the RNAseq, DNA methylation and HiC array data are deposited for public access.

Reviewer #3: In this study the authors report on the potential effects of the gene on lung function as well as evidence of mediation via DNA methylation. While a lot of information is presented in support of these findings, I have some methodological concerns regarding the control for asthma status and mediation analysis. Furthermore I am not sure the evidence presented support the potential role attributed to the PPP1R13B gene as a potential part of a causal mechanism.

Specific comments:

The motivation in the introduction (and the rest of the manuscript) alludes lung development, however the chosen spirometry parameters are FEV1 and the FEV1/FVC. Why was not FVC considered as an outcome? FEV1 and especially the FEV1/FVC ratio are parameters indicative primarily of airway obstruction/restriction. Decreased FVC would likely be a better indication of impaired lung development than at least the FEV1/FVC ratio.

The adjustment for asthma status particularly for the potential genetic effects is problematic. It is mentioned as part of a set of 'confounding factors', however it the structural sense of confounding it is not one. It can however be considered as an intermediate as it is likely caused by some genetic factors and in turn affects the outcome (lung function parameters). Its adjustment therefore is problematic as it will remove part of the potential effect of exposure, but also introduce the potential for collider bias if unmeasured common causes of asthma status and lung function exist. The two-stage approach for GWAS using the normalized residuals of the outcome does not protect from this issue. It was reassuring to see that the reported effect for the highlighted gene remained even without adjustment for asthma, but adjustment may have obscured effects of other genes and would also cause problems in the mediation analysis in quantification of direct and indirect effects.

The mediation results presented here do not refer to any of the assumptions required for mediation analysis or the assumptions under which mediated effects are estimated. Mediation analysis requires exchangeability assumptions that essentially require no unmeasured exposure-outcome, exposure-mediator and mediator-outcome confounding. The latter assumption in particular is the more problematic one as in this case the only other common cause of methylation and outcome considered is tobacco exposure (presumably from current SHS exposure). Any other environmental risk factor affecting both DNA methylation at the outcome would mean this assumption fails. Importantly the authors have provided enough evidence of mediation even in light of these limitations: The meQTL results for exposure-mediator relationship are in theory not thought to be very prone to confounding, while the instrumental variable approach for the mediator-outcome relationship is considered valid even in the presence of unmeasured confounding between methylation and the outcome if the instrument is valid. Both these relationships have a causal interpretation and together are sufficient to indicate mediation. The quantification of direct and indirect effects, however, is dependent on the aforementioned assumptions that are not even mentioned let alone addressed in any way. Proper quantification of direct and indirect effects should also take into account potential interactions between exposure and mediator if any exist, which I don't think the ROBMED package does.

Furthermore, since indirect effects are reported it is very difficult to compare to the total effects (assuming they are accurate if the assumptions above hold). Based on the betas reported, the indirect effect is greater than the total effect which would mean that the direct effect is in the opposite direction. Based on my understanding of the methods however, the betas aren't comparable (one is based on normalized residuals of the outcome, the other is FEV1 predicted untransformed). It the results are to be kept then it would greatly benefit the reader to see them in a scale that is comparable.

Identifiability assumptions and especially the issue of exchangeability also affects the issue of interactions. Identification of interactions in epidemiology both implies the presence of causal effects of the two (or more) variables of interest potentially interacting to jointly affect an outcome, but also that the causal effects of both of exposures are identifiable. This becomes an issue with (secondhand) smoking exposure especially as no socioeconomic indicators and co-exposures are adjusted for.

The focus on the PPP1R13B gene as a part of a potential causal mechanism for decreased lung function is problematic in the sense that it is not associated with the outcome of interest. If it was indeed part of the causal mechanisms then it should be associated with lung function parameters, while the proposed roles in the discussion would also lead us to expect that it should be associated with the outcome. The association with the risk alleles and DNA methylation alone does not necessarily implicate this gene on effects on lung function as the authors assert in the title and the discussion. In this sense the importance of this particular finding is overstated in the article.

Lastly, since most of the remaining analysis beyond the GWAS is only performed in URECA it would be interesting to see if the primary finding for TDRD9 persists in the URECA sample alone. A lot of the additional results are only based on URECA participants and even if the effect is still present but with a different magnitude it makes it difficult to compare total and indirect effects coming from only moderately overlapping samples.

Minor comment:

The authors use the general term of "smoking exposures" though given the age of the children in URECA this is likely secondhand smoke exposures. This should be better specified in the text.

**Have all data underlying the figures and results presented in the manuscript been provided?**

Reviewer #1: None

Reviewer #2: **No: **WGS data phs002921.v1.p1 cannot be found on dbGaP. There is no information about where the RNAseq, DNA methylation and HiC array data are deposited for public access.

Reviewer #3: Yes

PLOS authors have the option to publish the peer review history of their article (what does this mean?). If published, this will include your full peer review and any attached files.

Reviewer #1: No

Reviewer #2: No

Reviewer #3: No

---

## [Decision Letter · Decision Letter 1]

7 Dec 2022

Dear Dr Dapas,

Thank you very much for submitting your Research Article entitled 'Multi-omic association study identifies DNA methylation-mediated genotype and smoking exposure effects on lung function in children living in urban settings' to PLOS Genetics.

The manuscript was fully evaluated at the editorial level and by independent peer reviewers.  While two reviewers are satisfied, the third reviewer has raised a few minor points, which we hope that you can address in a revision.

Yours sincerely,

Wei Pan

Guest Editor

PLOS Genetics

Hua Tang

Section Editor

PLOS Genetics

Reviewer's Responses to Questions

**Comments to the Authors:**

Reviewer #1: authors have addressed all my concerns, well done!

Reviewer #2: This reviewer thanks the authors for addressing all the comments.

Reviewer #3: The authors have addressed comments from the initial review. I have some lingering concerns from the point of view of clarity/accuracy regarding some language used in the Discussion :

In the discussion (ln 338) the authors mention that PPP1R13B expression is not 'directly' associated with lung function. The word directly here is problematic in two ways. Terms 'direct' and 'indirect' in epidemiology are usually used to distinguish (causal) effects with respect to a specific mediator of interest which is not the case here as far as PPP1R13B expression and lung function are concerned. The presumed 'indirect association' alluded to here probably has to do with common causes of both PPP1R13B expression and lung function with respect to associated genes and DNA methylation, but absent any causal effect between PPP1R13B expression and lung function for which we have no evidence here, any such association would be by definition non-causal. However, since no association (causal/mediated, or non-causal) between PPP1R13B expression and lung function was reported here I think the sentence should be phrased as "PPP1R13B expression in airway epithelial cells at age 11 was not associated with lung function or urine cotinine levels in the URECA children, but ..." with the word directly entirely omitted.

In the same page the beginning of the last paragraph (lns 356-357) which reads "The correlations of FEV1 and rs10220464 with cg03306306 methylation and PPP1R13B expression in NECs..." reads as if FEV1 and rs10220464 and are correlated with both cg03306306 methylation and PPP1R13B expression in NECs, however this is not true based on the reported results, as FEV1 was not correlated with PPP1R13B expression in either NECs or PMBCs. This sentence should be restructured for clarification.

In the Discussion it is also stated the paragraph regarding interactions also can be misleading and lacks clarity. The paragraph begins (ln 372) by stating "If increased methylation at cg03306306 leads to lower FEV1 in response to environmental stressors, one might expect to see interaction effects with smoking exposure on FEV1". It is not clear what interaction is referred to here. One between methylation and smoking exposures or one between genes and smoking exposure? Regardless, I fear that the sentence is misleading in the sense that evidence of an established relationship between an exposure and mediator or two exposures sharing the same mechanism of effect as is the presumed case here with genetics and smoking exposures both affecting lung function through DAN methylation is by no means enough evidence to lead us to expect the presence of an interaction. The authors are correct in checking for any interaction, however the phrasing as is implies that the evidence at hand would lead us to believe that there might be an interaction, but this is not methodologically sound reasoning and should be rephrased.

**Have all data underlying the figures and results presented in the manuscript been provided?**

Reviewer #1: None

Reviewer #2: Yes

Reviewer #3: Yes

PLOS authors have the option to publish the peer review history of their article (what does this mean?). If published, this will include your full peer review and any attached files.

Reviewer #1: No

Reviewer #2: No

Reviewer #3: No

---

## [Editor Report · Decision Letter 2]

23 Dec 2022

Dear Dr Dapas,

We are pleased to inform you that your manuscript entitled "Multi-omic association study identifies DNA methylation-mediated genotype and smoking exposure effects on lung function in children living in urban settings" has been editorially accepted for publication in PLOS Genetics. Congratulations!

Yours sincerely,

Wei Pan

Guest Editor

PLOS Genetics

Hua Tang

Section Editor

PLOS Genetics

Comments from the reviewers (if applicable):

**Data Deposition**

http://datadryad.org/submit?journalID=pgenetics&manu=PGENETICS-D-22-00663R2

**Press Queries**

---

## [Editor Report · Acceptance letter]

11 Jan 2023

PGENETICS-D-22-00663R2 

Multi-omic association study identifies DNA methylation-mediated genotype and smoking exposure effects on lung function in children living in urban settings 

Dear Dr Dapas, 

We are pleased to inform you that your manuscript entitled "Multi-omic association study identifies DNA methylation-mediated genotype and smoking exposure effects on lung function in children living in urban settings" has been formally accepted for publication in PLOS Genetics! Your manuscript is now with our production department and you will be notified of the publication date in due course.

With kind regards,

Zsofia Freund

PLOS Genetics

On behalf of:
